# Fibroblast growth factor signaling induces a chondrocyte-like state of peripheral nerve fibroblast during aging

Dragana Stefanovska [1,2], Eliza Sassu[1], Mehmet Tekman [1], Amirhossein Naghsh Nilchi [1,3], Severin Haider [1], Claudia Domisch[1], Madelon Hossfeld [1], Stefanie Perez-Feliz[2], Lauritz Miarka [4], Franziska Schneider-Warme [2,5], Sebastian J. Arnold [1,5], Marco Prinz [4,5,6], Björn Grüning [3], Sebastian Preissl [1,5,7,8] ✉ & Luis Hortells [1,2,9] ✉

During aging, peripheral nerves undergo structural and cellular changes that trigger loss of function, impair quality of life, and increase disease risk. During peripheral nerve aging there are cellular and molecular changes, such as increased extracellular matrix deposition. The mechanisms behind these aging-induced alterations remain unclear. Here, we profile mouse sciatic nerves using single nucleus transcriptomics and unravel changes in macrophage subtypes during nerve aging. Phagocytic macrophage numbers increase at the onset of aging, followed by higher numbers of chronic inflammatory macrophages. Based on ligand-receptor analysis, we predict that increased fibroblast growth factor (FGF) signaling from adipocytes activates a chondrocyte-like neural fibroblast state during peripheral nerve aging. Finally, we show that FGF2 induces the co-expression of the chondrocyte markers SOX9 and FOXC2 in senescent human perineurial fibroblast, that can be blocked with FGF1. In conclusion, our findings reveal some of the molecular mechanisms of peripheral nerve aging by FGF-regulated induction of a chondrocyte-like fibroblast state.

Aging of the peripheral nervous system (PNS) reduces nerve conduction velocity[1] and general function, consequently affecting target organs such as skeletal muscle, heart or intestine[2–4]. In consequence of this alteration of the PNS activity, body function is impaired, the risk of disease increases, and life expectancy and quality of life decrease. Importantly, while consequences of PNS aging like skeletal muscle debilitation or neuropathy[5,6], and a subsequent increase in sedentarism, have been described, the cellular and molecular changes behind these conditions remain unclear[7,8].

During aging, organs like the heart[9], lungs[10], or liver[11], can develop fibrosis due to fibroblast activation. Additionally, aging can also be associated with phenotypic changes in other cell types, such as the transition of vascular smooth muscle cells into osteoblast-like cells during vascular calcification[12]. Furthermore, increased fat versus

[1]Institute of Experimental and Clinical Pharmacology and Toxicology, Faculty of Medicine, University of Freiburg, Freiburg, Germany. [2]Institute for Experimental Cardiovascular Medicine, University Heart Center Freiburg · Bad Krozingen, University of Freiburg, Freiburg, Germany. [3]Bioinformatics Group, Department of Computer Science, Albert-Ludwigs-University Freiburg, Freiburg, Germany. [4]Institute of Neuropathology, Faculty of Medicine, University of Freiburg, Freiburg, Germany. [5]CIBSS – Centre for Integrative Biological Signalling Studies, University of Freiburg, Freiburg, Germany. [6]Center Brain Research and Advancements In Neuroimmunology (BRAIN), Faculty of Medicine, University of Freiburg, Freiburg, Germany. [7]Institute of Pharmaceutical Sciences, Pharmacology & Toxicology, University of Graz, Graz, Austria. [8]Field of Excellence BioHealth, University of Graz, Graz, Austria. [9]Cardiovascular Research Group, Department of Medical Biology, Faculty of Health Science, UiT-The Arctic University of Norway, Tromsø, Norway. ✉e-mail: sebastian.preissl@uni-graz.at; sebastian.preissl@pharmakol.uni-freiburg.de; luis.hortells@uit.no

muscle tissue growth[13] and systemic inflammation[14] are well-established hallmarks of aging. In this direction, an increased deposition of chondral proteoglycans in the sciatic nerves from old mice has been reported recently[15], but the mechanisms and how neural fibroblasts might be involved in this extracellular matrix (ECM) remodeling are yet to be clarified.

Interestingly, fibroblast growth factor (FGF) signaling is implicated in age-related intramuscular fat deposition[16], fibrosis[17], and vascular calcification[18], but the precise mechanisms related to the activation of the FGF pathway during aging are not well described. In addition, FGF signaling is known to induce chondrocyte differentiation[19], but its role in the aging of the PNS remains unknown.

In peripheral nerves from young adults, SRY-box transcription factor 9 (SOX9) has been found to be present in endoneurial fibroblasts in healthy nerves[20], and also in satellite glial cells during reparative processes after nerve injury[21]. Besides this, SOX9 is also a chondrocyte marker, and it has been recently reported to play a role in several degenerative processes, including vascular calcification[22] or cardiac fibrosis[23]. However, whether there are changes in SOX9 levels during age-related proteoglycan deposition in the PNS remains unknown.

Single-cell or single-nucleus RNA-sequencing (snRNA-seq) are powerful tools to study cell populations and activation states, as well as potential cell-cell interaction patterns. While in the last 10 years this tool has helped to better understand the heterocellular composition of the peripheral nerves[24,25], and their cellular changes after nerve injury[20], we still lack of studies on peripheral nerve aging in order to understand age-associated changes of gene expression in the PNS.

In the present work, we use snRNA-seq to explore the dynamics of cell states and cellular composition of mouse sciatic nerves during aging. Macrophages (Mɸ) are one of the cell types that undergo profound changes during aging. While at the onset of aging, phagocytic Mɸ numbers are larger, at later time points, a chronic inflammation-Mɸ phenotype is more prevalent in the sciatic nerve. In addition, we found an age-related expansion of a fibroblast subpopulation that expressed a combination of chondrocyte-lineage markers (Sox9, Foxc2, and Cspg4), confirming these findings in human peripheral nerves. Re-analysis of a published dataset showed the presence of this subpopulation in nerves from P60 mice as a sub-part of the perineurial fibroblast cluster[25]. Using cultures of human perineurial fibroblasts, we predict that this transition of fibroblast states depends on FGF signaling from adipocytes to fibroblasts and show that FGF2 exposure leads to higher levels of SOX9 and forkhead box protein C2 (FOXC2). Notably, the fibroblast activation could be prevented by combining FGF1 with FGF2, offering an alternative perspective to promote healthy aging. In summary, we have found that FGF2 signaling promotes age-related PN degeneration via activation of a chondrocyte-like state of neural fibroblasts, that can be blocked with FGF1.

## Results

### Sciatic nerve snRNA-seq reveals age-related signatures in different cell types

Strong changes have been observed in the sciatic nerve from mice of 20 months old and older compared to those from young mice[15]. To interrogate whether these changes happened at an earlier time point, we studied two previously defined hallmarks of peripheral nerve aging[26]: loss of nerve fibers and loss of Schwann cells in 2-3, 15-16, and 20-30 months old mice. We observed that the relative numbers of S100B + Schwann cells were reduced already in 15-16 months old mice while SOX10 + Schwann cells were only significantly reduced in the 20-30 months old group (Supplementary Fig. S1a–c) Interestingly, we found that while Sox10 is expressed in myelinating and non-myelinating Schwann cells (Supplementary Fig. S1d), S100b is mainly expressed in myelinating Schwann cells (Supplementary Fig. S1e, f). Therefore, it is likely that myelinating Schwann cells are affected earlier

by aging. In addition, statistically significant differences in sympathetic nerve fibers were observed only when comparing young to old mice (20-30 mo, Supplementary Fig. S1g, h), and we did not find significant, age-related differences in relative myelin-positive area using a fluorescent myelin probe (Supplementary Fig. S1g, i). Then, we examined peripheral nerve cellular diversity at different time points of the mouse lifespan using snRNA-seq of sciatic nerves from 2-3, 15-16 and 20-30 months old mice (Fig. 1a). To ease the access to these results, we created an online public platform (go to data availability statement in methods). In our initial assessment, we detected 19 cell clusters (Fig. 1b–d and Supplementary Fig. S2a). We annotated 18 out of 19 clusters based on previous works on young (2-3 months old) mice, (Fig. 1c and Table 1)[24,25]: Endoneurial fibroblasts (Enpp2 +, Col2a1 +), adipose tissue cells (Fabp4 +, Car3 +, Cdo1 +), moto-neuron-associated myelinating Schwann cell (mSC, Pmp2 +, Cldn14 +, Mag +), perineurial fibroblasts (PnFbs, Slc2a1/Glut1 +, Itgb4 +), macrophages (Mɸ, Ptprc1 +, Csf1r +, Adgre1 +, Bank1-, Cd247-), high-Plp1 mSC (Plp1$^{high}$, Mag +), epineurial cells (Pi16 +, Pcolce2 +), B-cells (Ptprc1 +, Bank1 +, Cd247-), endothelial cells 1 and 2 (EC1, EC2, Pecam1 +, Prox1-), Nerve-associated EC (Pecam1 +, Cldn5 +, Prox1-), Low-Plp1 mSC (Plp1$^{low}$, Mag +), non-myelinating SC (nmSC, Ncam1 +, Slc35f1 +) T- and NK-cells (Ptprc1 +, Bank1-, Cd24 +, Cd3g +, Nkg7 +), lymphatic endothelial cells (Pecam1 +, Prox1 +), pericytes (Tgln +, Acta2 +, Mcam-), and vascular smooth muscle cells (VSMC, Tgln +, Acta2 +, Mcam +, Fig. 1e). In addition, we detected a previously undescribed population of fibroblasts that expressed a combination of chondrocyte-lineage markers (Sox9, Foxc2, and Cspg4), as well as low levels of the perineurial fibroblast marker Slc2a1/Glut1. Next, we explored whether there is an enrichment of any population based on age. We found that all cell populations were present at all time points. Adipose tissue cells were found in higher numbers in nerves from 20-30 months old mice, and chondrocyte-like state fibroblast were more prevalent in mice older than 15 months (Fig. 1d, f and Supplementary Fig. S2b, c). Relative numbers of B-cells, endothelial cells (EC), and endo-, peri-, and epineurial fibroblasts relative numbers were not different between age groups (Fig. 1f and Supplementary Fig. S2d–g). These results corroborate previous findings in sciatic nerves from young adult mice and provide some insight into age-related cell population changes of the PNS.

### Changes in sciatic nerve Mɸ profile during aging

One of the hallmarks of aging is the development of a global inflammatory state, also known as inflammaging, which is reflected by Mɸ activation in peripheral nerves[14]. To explore possible age-related expression changes, we re-clustered the Mɸ cluster and annotated six second-level clusters: endoneurial, epineurial, circulating-derived phagocytic, chronic inflammation, circulating-derived_1, and circulating-derived_2 Mɸs (Fig. 2a). At least three of these clusters appeared enriched at different age points when we interrogated the time-point of origin of each nucleus in the UMAP (Fig. 2b and Supplementary Fig. S3a–c). We annotated these second level clusters based on both the expression of key marker genes[27–30] (Fig. 2c), and on their biological function (Supplementary Fig. S3b–g). Interestingly, the phagocytic and chronic inflammation Mɸs were enriched in sciatic nerves from 15-16 and 20-30 months old mice, respectively (Fig. 2b, d and Supplementary Fig. S3a). Moreover, in the sciatic nerves of young mice, Mɸs express more resident (Cx3cr1) and steady state genes (Mrc1 and Csf1r), while those from 15-16 months old mice express more circulating-derived (Ccr2, Cd74, and Hspa5) and phagocytic phenotype related genes (Cd44, C1qa, and Grn). Finally, in sciatic nerve Mɸs from 20-30 months old mice, we found increased expression levels of chronic inflammation-related genes like Tgfβr1, and Kynu (Fig. 2e).

To test our findings at the protein level, we performed immunofluorescence and posterior cell counting of sciatic nerves using antibodies against the bone marrow-derived cell marker CD45, pan-Mɸ

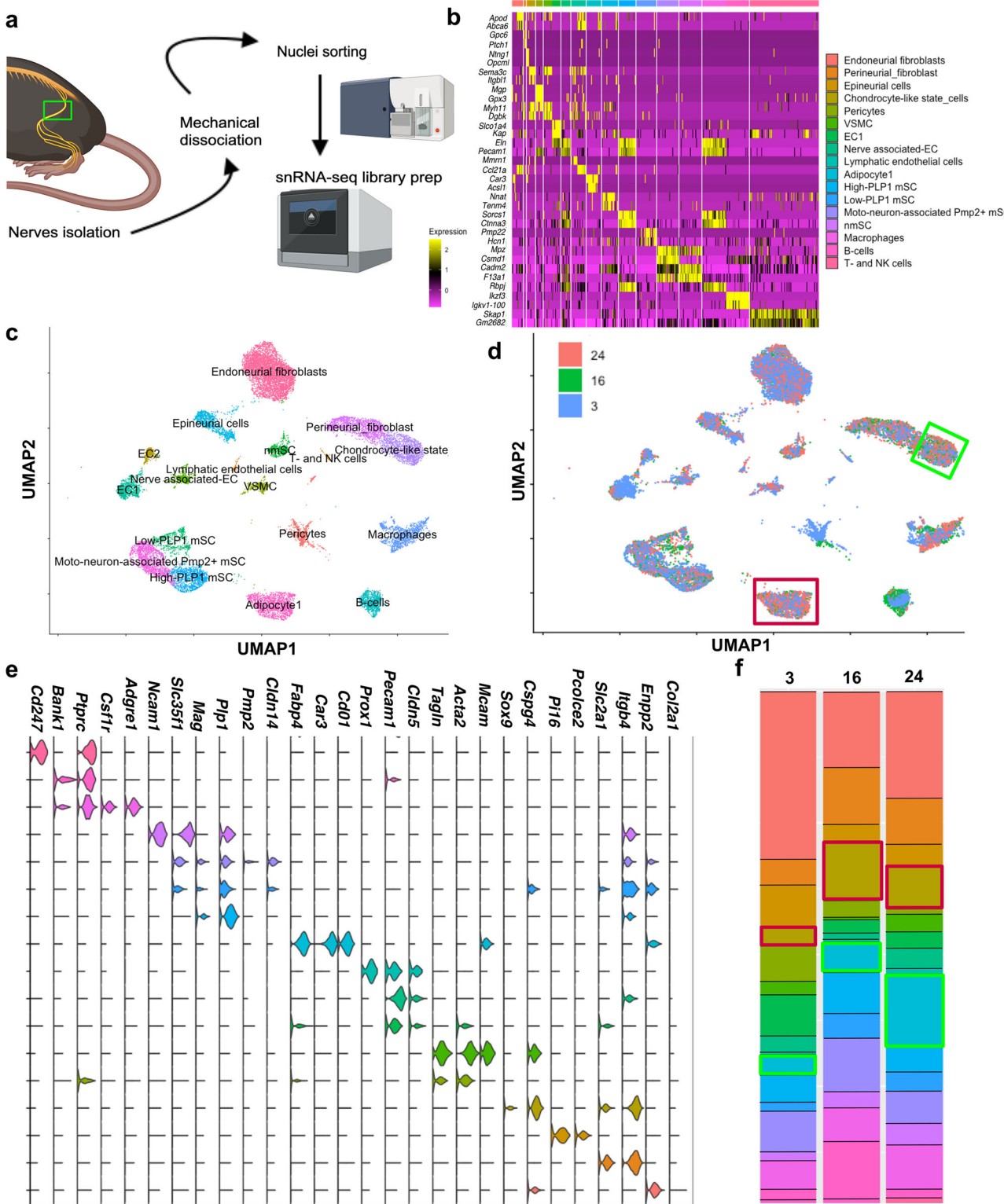

**Fig. 1 | First level clustering of snRNAseq from sciatic nerves from 2-3, 15-16, 20-30 months old mice. a** Schematic diagram of the single nuclei isolation, FACS, and library preparation. Elements were created in BioRender. Preissl, S. (2025) https://BioRender.com/2nbvklt. **b** Heatmap of the 2 top differentially expressed genes in each of the 19 first level clusters: Endoneurial fibroblasts (*Abca6*, *Apod*), Perineurial fibroblasts (*Ntng1*, *Gpc6*), Epineurial cells (*Opcml*, *Sema3c*), Chondrocyte-like state cells (*Ptch1*, *Itgbl1*), Pericytes (*Mgp*, *Gpx3*), VSMC (*Myh11*, *Dgkb*), EC1 and 2 (*Slco1a4*, *Kap)*, Nerve associated EC (*Pecam1*, *Eln*), Lymphatic EC (*Mmrn1*, *Ccl21a*), Adipose tissue 1 (*Car3*, *Acsl1*), Adipose tissue 2 (*Nnat*, *Tenm4*), Low-Plp1 mSC (*Sorcs1*, *Ctnna3*), Moto-neuron associated mSC (*Pmp22*, *Slit2*), High-Plp1 mSC (*Hcn1*, *Mpz*),

nmSC (*Csmd1*, *Cadm2*), Macrophages (*F13a1*, *Rbpj*), B-cells (*Ikzf3*, *Igkv1*-110), T- and NK cells (*Skap1*, *Gm2682*). Cluster color code is maintained through the figure. *n* = 3, biological replicates. **c, d** UMAP embedding of 19,175 high-quality nuclei color-coded by cell type (**c**) or age (**d**) red box highlights adipocytes and green box highlights chondrocyte-like state fibroblasts. **e** Violin plots illustrating different signature genes expressed in 19 clusters. **f** Relative percentage of clusters across ages. Adipose tissue cells and chondrocyte-like state cells relative presence increases with age. Green boxes mark the chondrocyte-like state fibroblasts, and red boxes mark adipocyte clusters.

marker CD68, the steady state marker MRC1, the phagocytic Mφ marker GRN, and the chronic inflammation Mφ marker TGFβR1 (Fig. 2f and Supplementary Fig. S3h). While no difference in the total number of CD45 + or CD68 + Mφs was observed at the different time points (Fig. 2g and Supplementary Fig. S3h), MRC1 +/CD45 + endoneurial steady state-Mφs were significantly more prevalent in sciatic nerves from 2-3 months old mice (Fig. 2h), GRN +/CD68 + phagocytic Mφs in sciatic nerves from 15-16 and 20-30 months old mice (Fig. 2i), and TGFβR1 +/CD45 + Mφs in the 20-30 months old sciatic nerve samples (Fig. 2j), confirming our initial findings based on gene expression. We also interrogated whether Mφs were phagocyting myelin by using a fluorescent tag for myelin combined with CD68 as Mφ marker. We detected higher numbers of Mφs with intracellular myelin in sciatic nerves from 15-16 and 20-30 months old mice compared to young mice (Supplementary Fig. S3h, h'). These findings point to a dynamic phenotype of Mφs during aging, with an activation of a phagocytic phenotype at 15-16 months old and a chronic inflammatory Mφ phenotype at 20-24 months old mice, thus supporting the development of a pro-inflammatory environment in PNS aging.

## During aging, nerve fibroblasts can acquire a chondrocyte-like state

Our initial observations pointed to the presence of a chondrocyte-like state fibroblast cluster, whose numbers increase with age (Fig. 1d, f). During aging there is a well-described development of fibrosis in organs like the heart or the lung, and calcification in other soft tissues like the heart and blood vessels. Therefore, we explored age-related gene expression changes in the different fibroblast populations of the sciatic nerve. At the second level of clustering, we annotated two endoneurial, an epineurial, and a perineurial fibroblast cluster, in addition to the previously annotated chondrocyte-like state fibroblast cluster (Fig. 3a). All the clusters were present across the different time points, but the chondrocyte-like state fibroblasts were more prevalent in sciatic nerves from 15 months old mice or older (Fig. 3b). Using the list of differentially expressed genes, we explored the biological function of these chondrocytic cells using Enrichr[31] (Fig. 3c). The top hit was skin morphogenesis with genes like *Col1a1* or *Col1a2* indicating a possible pro-fibrotic phenotype, and interestingly, the second and third predicted functions were chondrocyte development and positive regulation of chondrocyte differentiation, which include several Sox transcription factors such as *Sox9*, *Sox6* and *Sox5*, suggesting a chondrocyte-like state of these fibroblasts. Next, we found that the putative chondrocyte-like state fibroblast cluster was the only one with co-expression of the chondrocyte-like genes *Cspg4*, *Sox9*, and *Foxc2* (Fig. 3d). Furthermore, we interrogated at what time point these chondrocytic fibroblast express higher levels of these markers and found their expression increased in the late time points (Fig. 3e). We further tested these results at the protein level by immunofluorescence (Fig. 3f). We found that both SOX9+ cell density and CSPG4 area were significantly increased at the 20-30 month time point (Fig. 3g, h).

Finally, we explored whether any sciatic nerve fibroblast population was more likely to activate this chondrocyte-like state. For this, we performed pseudotime analysis using PnFbs, epineurial, and endoneurial fibroblast as origin nodes and chondrocyte-like state fibroblasts as endnode (Fig. 3i). Interestingly, the more probable origin based on this analysis were PnFbs and the least probable origin were endoneurial fibroblasts. To explore this question in vivo, we quantified the number of SOX9 + epineurial fibroblasts (detected using the marker KCNC2) and PnFbs (detected using the marker GLUT1, and their perineurial localization) in 20-30 months old mice sciatic nerves (Fig. 3j). In agreement with the pseudotime analysis, we found that the number of SOX9 + PnFbs was significantly higher than the epineurial fibroblasts (Fig. 3k). Together, our data confirms the presence of a chondrocyte-like state of fibroblasts, that has high expression levels of

**Table 1 | Marker genes used to identify first level clusters**

| Cell type | Marker genes |
|---|---|
| Endoneurial fibroblasts | *Enpp2 +, Col2a1 +* |
| Adipose tissue 1 | *Fabp4 +, Car3 +, Cdo1 +* |
| Moto-neuron associated Pmp2+ mSC | *Pmp2 +, Cldn14 +, Mag +* |
| Perineurial fibroblasts | *Slc2a1 +, Itgb4 +, Ptprc-* |
| Chondrocyte-like state cells | *Sox9 +, Cspg4 +, Slc2a1 +* |
| Macrophages | *Ptprc +, Csf1r +, Adgre1 +, Bank1-, Cd247-* |
| High-Plp1 mSC | *Mag +, Plp1^{high}* |
| Epineurial cells | *Pi16 +, Pcolce2 +* |
| B-cells | *Ptprc +, Bank1 +, Cd247-* |
| EC1 | *Pecam1 +, Prox1-* |
| Low-Plp1 mSC | *Mag +, Plp1^{low}* |
| nmSC | *Ncam1 +, Slc35f1 +* |
| Nerve-associated EC | *Pecam1 +, Cldn5 +, Prox1-* |
| Adipose tissue 2 | *Fabp4 +, Car3 +, Cdo1 +* |
| T- and NK- cells | *Ptprc +, Bank1-, Cd247 +, Cd3g +, Nkg +* |
| Lymphatic endothelial cells | *Pecam1 +, Prox1 +* |
| Pericytes | *Tagln +, Acta2 +, Mcam-* |
| VSMC | *Tagln +, Acta2 +, Mcam +* |
| EC3 | *Pecam1 +, Prox1-* |

CSPG4, and that most likely is originated from PnFbs in the sciatic nerve of aged mice.

## During nerve aging there are increased numbers of FGF2+ adipocytes

To better understand the signaling mechanisms associated with the activation of the chondrocyte-like state in neural fibroblasts, we used CellChat[32] to predict possible cell-cell interactions in the nerves (Fig. 4a). One of the first-level adipocyte clusters showed increased predicted interactions (Fig. 4b) and interaction strength (Fig. 4c) with PnFbs and chondrocyte-like state fibroblasts. FGF signaling was an interesting candidate pathway (Fig. 4d, e) since FGF in general and FGF2 in particular are known to induce *Sox9* expression and chondrocyte differentiation[33]. In addition, FGF2 directly interacts with CSPG4 based on the STRING database[34], which we also found increased in sciatic nerves from mice 15 months old or older (Figs. 3f, 4f). Therefore, we explored whether FGF signaling from adipocyte could activate a chondrocyte-like state fibroblast. First, we found that *Fgf2* was higher expressed in adipocytes compared to other clusters (Supplementary Fig. S4a), and the highest Cell-Chat scores for the FGF pathway in adipocytes were *Fgf2-Fgfr1* and *Fgf2-Fgfr2* (Supplementary Fig. S4b). In nerves from 20-30 months old mice, we detected higher expression of *Fgf2* and *Fgf1* in adipocytes as well as higher expression of *Fgfr1* and *Fgfr2* expression in chondrocyte-like state fibroblasts (Supplementary Fig. S4c–e). In agreement with Cell-Chat, we found higher *Fgf* ligand and receptor expression in nerves from 20-30 months old mice. Next, we gathered the adipocytes from the first level clustering, and re-clustered them into three second level subclusters (Fig. 4g). *Fgf2*, *Fgf1*, *Fgf10* were highly expressed in adipocyte cluster 2, while adipocyte cluster 1 only showed moderate expression of *Fgf2* (Fig. 4h). The relative presence of the different adipocyte clusters did not show major differences between age groups, with only some increased presence of adipocyte_3 in the 15-16 months old age group (Fig. 4i and Supplementary Fig. S3f). On the other hand, global and individual cluster expression of *Fgf7* and *Fgf10* in adipocytes was increased starting at 15-16 months old, and *Fgf2* and *Fgf1* were only increased in the 20-30 months old age group (Fig. 4j and

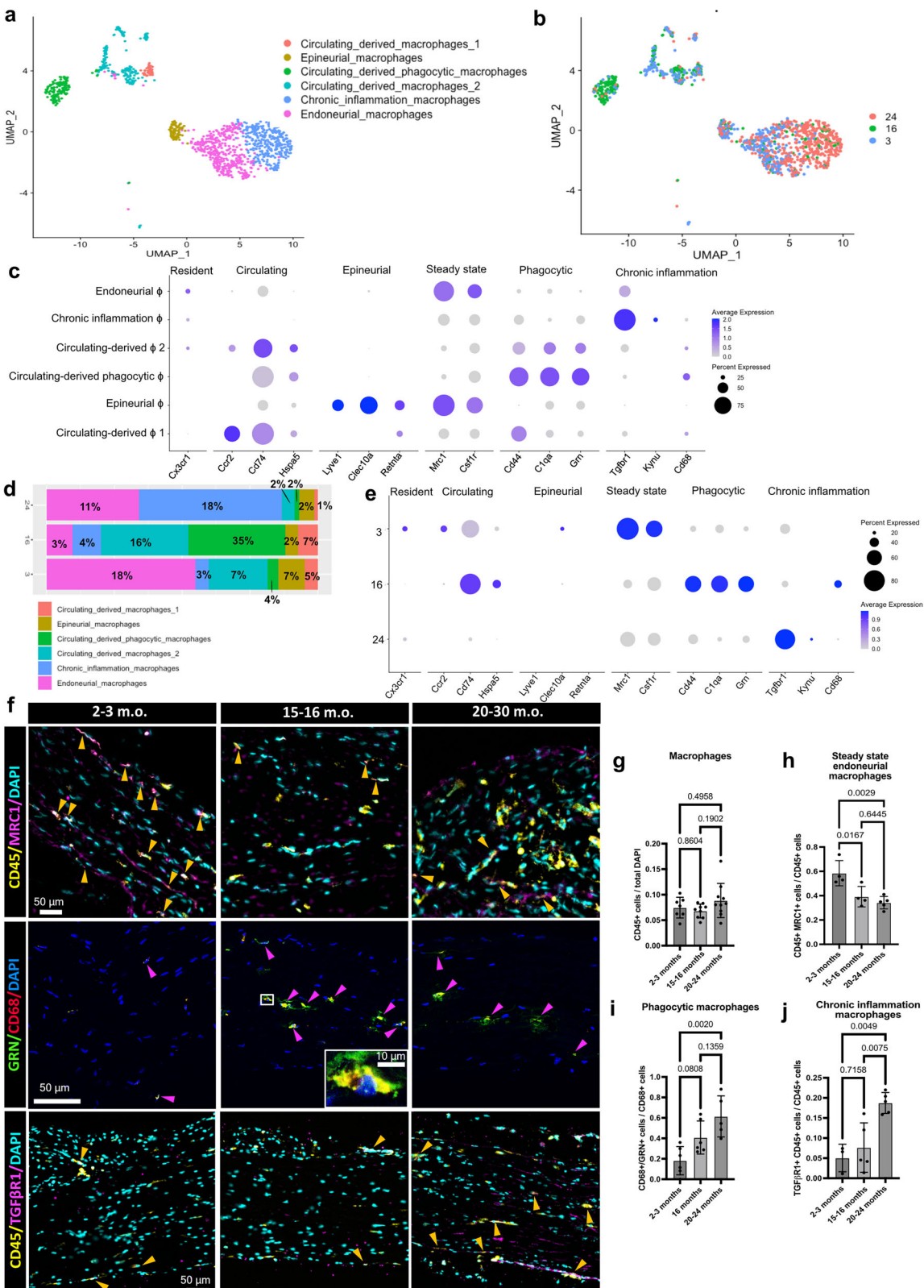

Supplementary Fig. S3g). In addition to increased levels of FGF signaling, we also found increased expression of adipocyte markers like *Fabp4* or *Adipoq* in sciatic nerves from 20-30 months old mice (Fig. 4k), corresponding with the increased number of adipocyte nuclei numbers (Supplementary Fig.S2b). Next, we counted adipocyte numbers by performing FABP4 immunostainings (Fig. 4l), finding significantly

higher density of adipocytes (Fig. 4m) and FGF2-expressing adipocytes (Fig. 4n) in the nerves from aged mice. These adipocytes were detected, as expected, mainly in the epi- and peri-neurium, making local adipocyte-human(h)PnFbs interaction feasible (Fig. 4l). Together, these findings indicate an increased density of FGF-expressing adipocytes in the epi-/perineurium of sciatic nerves from aged mice.

**Fig. 2 | PNS Mφ have different phenotypes during aging. a**, **b** UMAP embedding of 1180 macrophage nuclei from sciatic nerves color-coded by subtype (**a**) or age (**b**). **c** Dot plot highlighting the marker genes used to identify the macrophage subtypes from A. Marker genes can be found in Table 4. **d** Relative percentage of Mφs subtypes across ages. Steady state epi- and endoneurial Mφ, circulating-derived phagocytic Mφ, and chronic-inflammation Mφ were more prevalent in sciatic nerves from 2-3, 15-16, and 20-30 months old mice, respectively. **e** Dot plot showing the expression levels of the marker genes used to identify the macrophage clusters, at the different time points. Markers for resident steady state Mφ (*Cx3cr1, Mrc1, Csf1r*) are more expressed at 2-3 months old, those for circulating-derived phagocytic Mφ (*Cd74, Hspa5, Cd44, C1qa, Grn, Cd68*), are more expressed at 15-16 months old, and those belonging to chronic inflammation (*Tgfβr1, Kynu*) are more expressed in the 20-30 months old samples. **f** Representative immunofluorescence images of sciatic nerve samples show increased CD45 + /MRC1+ Mφs in nerves from 2-3 months old mice, CD68 + /GRN+ Mφs in nerves from 15-16 and 20-30 months old mice, and CD45 + /TGFβR1+ in nerves from 20-30 months old mice. Yellow or pink arrowheads point to CD45 + /MRC1+ or CD68 + /GRN+ or CD45 + / TGFβR1+ cells. **g** Quantification of the number of CD45+ cells relative to the total number of cells. *n* = 7 - 9 - 10, biological replicates. **h** Quantification of the number of CD45 + /MRC1+ cells relative to the total number of CD45 + cells. *n* = 4 - 4 - 5 biological replicates. **i** Quantification of the number of CD68 + / GRN+ cells relative to the total number of CD68+ cells. *n* = 6 - 6 - 5. **j** Quantification of the number of CD45 + /TGFβR1 + cells relative to the total number of CD45 + cells. *n* = 3 - 5 - 5, biological replicates. Data are presented as mean values +/− SD. Source data for panels is provided as a Source Data file.

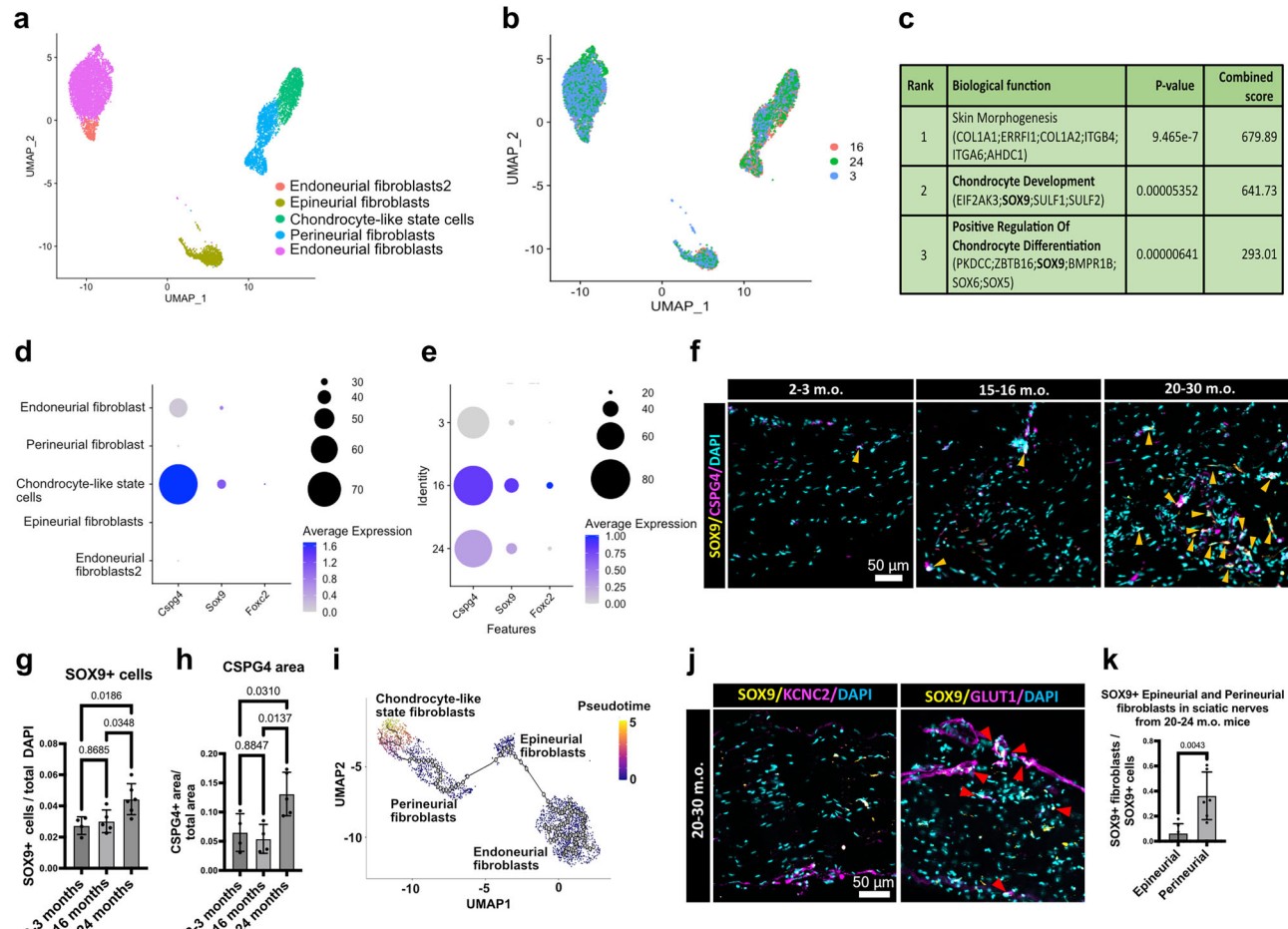

**Fig. 3 | PnFbs can activate a chondrocyte-like state during aging. a**, **b** UMAP embedding of 8495 fibroblasts colored by (**a**) cell type and (**b**) age. **c** Gene ontology analysis of biological functions of the chondrocyte-like cells based on the combined score. Main functions are related to collagen deposition (skin morphogenesis) and chondrocyte-differentiation (chondrocyte development and positive regulation of chondrocyte differentiation). **d** Dot plot of the chondrocytic genes *Cspg4, Sox9*, and *Foxc2* across the different neural fibroblast populations. The three genes are enriched almost exclusively in the chondrocyte-like state fibroblast cluster. **e** Dot plot of the chondrocytic genes *Cspg4, Sox9*, and *Foxc2* in chondrocyte-like state fibroblasts at different time points. Higher expression of all the markers is found in 15-16 and 20-30 months old samples. **f** Representative immunofluorescence images of sciatic nerve samples show increased SOX9+ cells and CSPG4+ area in sciatic nerves from 20-30 months old mice. Yellow arrows point to SOX9+ cells. **g** Quantification of the number of SOX9+ cells relative to the total number of cells. *n* = 4 - 5 - 6, biological replicates. **h** Quantification of the CSPG4 area relative to the total nerve area. *n* = 4 - 4 - 5, biological replicates. **i** Monocle pseudotime trajectory and scores overlaid on the UMAP projection of endo-/peri-/epi-neural fibroblasts from the 2-3 months old sciatic nerves, and chondrocyte-like fibroblasts from the 15-16 and 20-30 months old sciatic nerves. **j** Representative immunofluorescence images of sciatic nerve samples show more density of SOX9 + /GLUT1 than SOX9 + /KCNC2 + cells in sciatic nerves from 20-30 months old mice. Red arrows point to SOX9 + /GLUT1 + . **k** Quantification of the number of SOX9 + epineurial (KCNC2 + ) or perineurial (GLUT1 + ) fibroblasts relative to the total number of cells in sciatic nerves from 20-30 months old mice. *n* = 6, biological replicates. Data are presented as mean values +/− SD. Source data for panels is provided as a Source Data file.

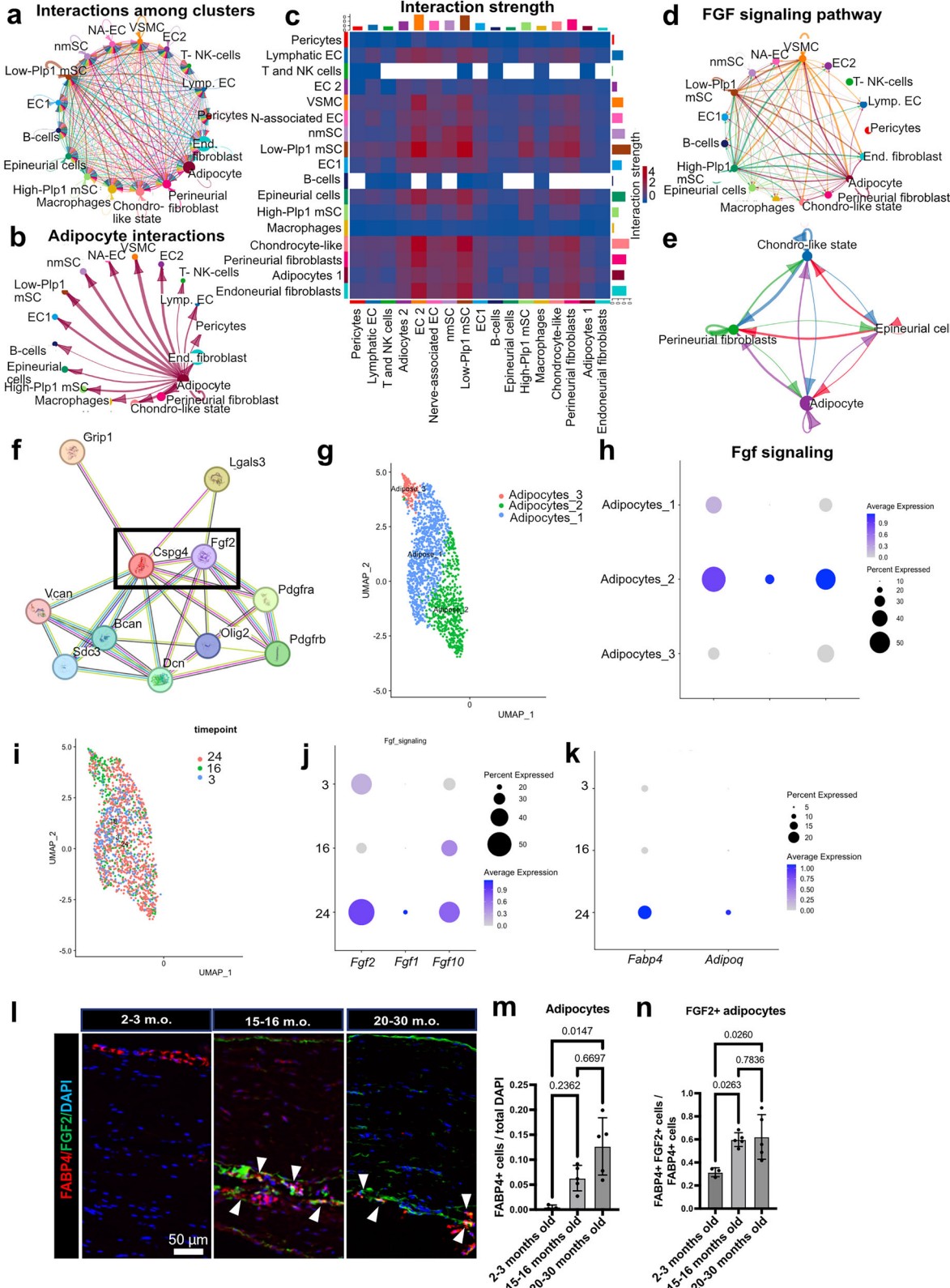

**FGF2 does not induce a chondrocyte-like state in early passage human perineurial fibroblasts (hPnFbs)**

While FGF2 was higher expressed in sciatic nerves from old mice, our data indicated moderate expression also in those from 2-3 months old mice (Fig. 4j), with no chondrocyte-like state activation (Fig. 3e). Thus, we explored the effect of recombinant FGF2 on human perineurial fibroblasts (hPnFbs) in vitro. After 14 days of treatment with 3 different

concentrations of FGF2 (1, 10, and 100 ng/ml, Supplementary Fig. S5a), we did not detect increased cell cycling (EdU +, Supplementary Fig. S5b), SOX9 + (Supplementary Fig. S5c), or SOX9 + /FOXC2 + (Supplementary Fig. S5d) cells. In addition, no change in CSPG4 levels was detected upon FGF2 stimulation (Supplementary Fig.S6a, b). Interestingly, we found fewer positive nuclei for the DNA damage/ senescence marker gamma-H2A histone family member X (γH2AX) +

**Fig. 4 | During aging, FGF2-secreting adipocyte density is increased in the sciatic nerve. a** CellChat inferred ligand-receptor interactions between any pair of two cell populations in the sciatic nerves from 20-30 months old nerves. **b** CellChat inferred ligand-receptor interactions between adipocyte cluster 2 and any cell populations in the sciatic nerves from 20-30 months old nerves. **c** Heatmap showing the interaction strength between any pair of cell populations in the sciatic nerves from 20-30 months old mice. **d** The inferred FGF signaling pathway between any pair of two cell populations in sciatic nerves from 20-30 months old nerves. **e** The inferred FGF signaling pathway between Adipocyte clusters 1 and 2, Endo-/Peri-/Epi-neurial fibroblasts, and chondrocyte-like state fibroblasts in sciatic nerves from 20-30 months old nerves. **f** STRING protein-protein association network analysis for mouse CSPG4. FGF2 is in the first interaction level. **g** UMAP embedding of 1552 adipocytes. **h** Dot plot showing the expression levels of the FGF ligands *Fgf2*,

*Fgf1*, and *Fgf10* in the three adipocyte second-level clusters. **i** UMAP embeddings colored by age. Increased expression of the FGF ligands is detected in adipocyte cluster 2. **j** Dot plot of the FGF ligands *Fgf2*, *Fgf1*, and *Fgf10* in adipocytes at different time points. Increased expression of Fgf2 and Fgf1 in adipocytes is observed in the 20-30 months old age group. **k** Dot plot of the adipocyte marker genes *Fabp4* and *Adipoq* in sciatic nerves at different time points. Higher expression of all the markers is found in 20-30 months old samples. **l** Representative immunofluorescence images of sciatic nerve samples show more density of FABP4 + /FGF2+ adipocytes in sciatic nerves from 15-16 and 20-30 months old mice. White arrows point to FABP4 + /FGF2+ adipocytes. **m, n** Quantification of the number of FABP4 + and FABP4 + /FGF2 + (respectively) adipocytes in sciatic nerves from 2-3, 15-16, and 20-30 months old mice. N 3 - 3 - 5, biological replicates. Data are presented as mean values +/− SD. Source data for panels is provided as a Source Data file.

when hPnFbs were exposed to 10 ng/ml of FGF2. These results point to a potential protective role of FGF2 against DNA damage in young/healthy hPnFbs (Supplementary Fig. S5e, f).

## FGF2, but not FGF1, induces a chondrocyte-like state in senescent hPnFbs

One of the hallmarks of aging is cellular senescence, and senescent cells can react differently than "young" cells to signals[35]. In this line, we interrogated the presence of senescent cells in the PNS of aged mice by using the DNA damage/senescence marker γH2AX. We did this in combination with the mesenchymal marker Vimentin (VIM), in sciatic nerves from *Sox10*[CreERT2/+]*;R26*[tdt/+] where Schwann cells are traced with the red fluorescent protein TdTomato (Fig. 5a). Using 2-3 months old mice and 20-24 months old mice, we detected significantly increased density of γH2AX+ Schwann cells (Fig. 5b) and *Sox10*-/VIM+ mesenchymal cells (which includes all fibroblast populations of the sciatic nerve, Fig. 5b) in nerves from 20-24 months old mice when compared with those from young mice. Based on this, we aimed to induce senescence of hPnFbs by increasing the number of passages, and therefore their cell division cycles. Using increased γH2AX expression and reduced EdU signal as markers of senescence, we detected significant changes starting at passage 8 (Fig. 5c, d). Thus, we used passage 8 and 9 as a model of hPnFb senescence. In addition, we further confirmed a progressive global senescent state in sciatic nerves from 15 months old mice and older, based on increased expression of senescence-related genes (*Btg1*, *TrpS3*, *Cdh5*, and *Cul4a*, Fig. 5e) and reduced expression of proliferation related genes (*Sphk1*, *Cdk1*, and *Cdk2*, Fig. 5f).

Given the increased number of senescent fibroblasts during aging of peripheral nerves, we explored the effect of FGF2 in senescent hPnFbs, where we treated passage 8 and 9 hPnFbs with the same FGF2 doses as used on the healthy cells (Fig. 5g). While proliferation and individual expression of SOX9 remained unchanged (Fig. 5h, i), significantly increased numbers of chondrocyte-like hPnFbs defined by the co-expression of FOXC2+ and SOX9 + , were found upon stimulation with 100 ng/ml of FGF2 (Fig. 5j). Interestingly, FGF2 did not protect senescent hPnFbs against DNA damage (Fig. 5k, l), contrarily to what was observed in early passage hPnFbs (Supplementary Fig. S5f). In the same line, the expression of CSPG4 remained unchanged (Supplementary Fig. S6c, d). Together, this data indicates that high concentrations of FGF2 can activate the co-expression of FOXC2 and SOX9, but other signals might be behind CSPG4 increased expression in senescent hPnFbs.

As we have also observed low expression levels of FGF1 in adipocytes from old mice (Fig. 4j), we tested whether FGF1 can also induce chondrocyte protein expression in senescent hPnFbs. For this, we treated senescent hPnFbs with three different concentrations of FGF1 and its co-activator, heparin, for 14 days (Supplementary Fig. S7a). Interestingly, we found that FGF1 does not recapitulate our observed effect with FGF2 (Supplementary Fig. S7a–c). No significant differences

were found in SOX9+ (Supplementary Fig. S7b) or FOXC2 + /SOX9 + (Supplementary Fig. S7c) hPnFbs numbers. In addition, CSPG4 levels were also unchanged (Supplementary Fig. S7d, e). Given that in nerves from old mice some expression of FGF1 was also detected in adipocytes (Fig. 4j), we decided to interrogate the effects of a combined treatment with the three different doses of FGF1 and FGF2 on senescent hPnFbs (Supplementary Fig. S7f). The results from this experiment did not show any significant difference in proliferative (Supplementary Fig. S7g), SOX9 + (Supplementary Fig.S7h), or FOXC2 + /SOX9 + hPnFbs (Supplementary Fig. S7i), suggesting that FGF1 can block FGF2 activation of a chondrocyte-like state in hPnFbs. In addition, CSPG4 deposition was reduced (Supplementary Fig. S7j, k). In the sciatic nerves, the expression levels of *Fgf1* detected in adipocytes were quite low compared to *Fgf2* in our snRNAseq analysis (Fig. 4j). Therefore, to better mimic our findings in vivo, we decided to compare the effect of low concentrations of FGF1 in combination with high concentrations of FGF2 on hPnFbs (Supplementary Fig. S8a). Low concentrations of FGF1 were insufficient to block the FGF2 effects, with significantly more SOX9 + (Supplementary Fig. S8b) and SOX9 + /FOXC2 + (Supplementary Fig. S8c) hPnFbs after treatment. On the other hand, no change in CSPG4 expression was detected (Supplementary Fig. S8d, e). Together, this shows that in a more similar environment to what is observed during aging in vivo, FGF2 can activate co-expression of SOX9 and FOXC2, but higher FGF1 concentrations can block activation.

To better define committed cellular transdifferentiation contrarily to cell state activation, we interrogated whether the induction of FOXC2 and SOX9 expression after stimulation with FGF2 is reversible after retrieval of the FGF2 stimulus. In these experiments we started treating senescent hPnFbs for 14 days with the same concentrations of FGF2 previously described, followed by seven days with normal media without FGF2 (Fig. 5m). Interestingly, in the cultures treated with 100 ng/ml of FGF2, there were significantly less FOXC2 + /SOX9 + hPnFbs after retrieval than after the 14 days treatment (Fig. 5n). Surprisingly, CSPG4 level was also reduced after seven days of FGF2 withdrawal (Supplementary Fig. S6e, f). In summary, our data confirms that high levels of FGF2, but not FGF1, signaling can activate a chondrocyte-like phenotype in senescent hPnFbs in a reversible manner. Additionally, high doses of FGF1 seem to induce the opposite effect, reducing the basal number of FOXC2 + SOX9 + cells.

Finally, we interrogated whether the relative presence of SOX9+ cells and relative expression of FGF2+ were increased in human peripheral nerves from young (17-33 years old) compared to older donors (55−68 years old, Fig. 5o). We found that the nerve samples from older donors had significantly more FGF2 + , and SOX9 + /FGF2 + cells compared to the nerves from young donors. On the other hand, no statistical significance was found in the number of SOX9 + cells (Fig. 5p−r). Finally, we also found significantly lower relative nuclei numbers in the samples from older donors, supporting the loss of cell density reported in peripheral nerves from old mice[36] (Fig. 5s). Together, this

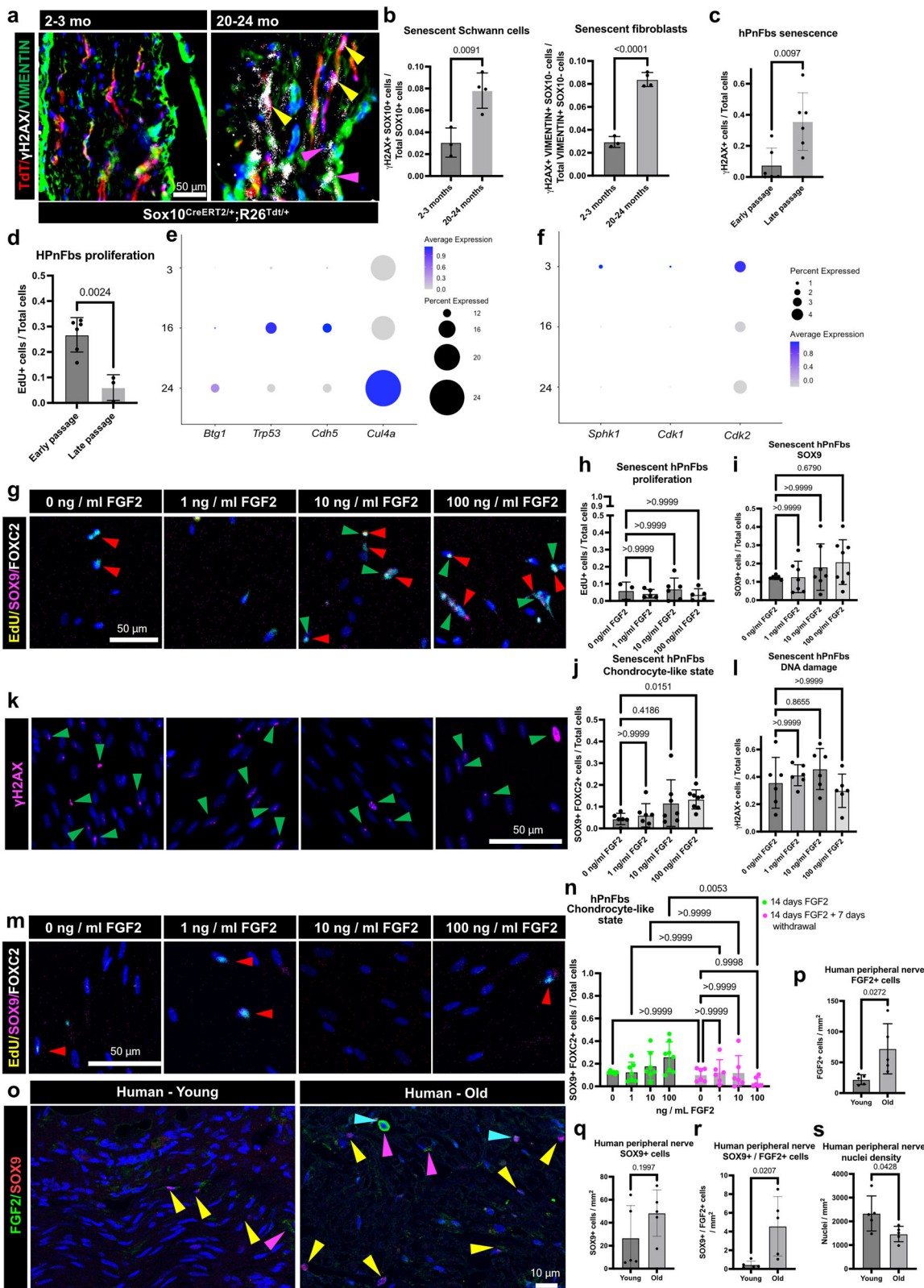

information suggests possible similarities between aging on murine and human peripheral nerves.

## Discussion

In this work, we have profiled cellular and transcriptional changes in sciatic nerves during mouse aging. We identified changes in macrophage subtypes and identified an age-related chondrocyte-like state in PNS fibroblasts. In addition, we have found that FGF2, possibly originating from adipocytes in the aging sciatic nerves, can activate this state in hPnFbs, and that high doses of FGF1 can block this process (Fig. 6).

We confirmed the diversity of cell populations previously observed in sciatic nerves from young mice, including the motorneuron-associated *Pmp2* + mSCs[24]. In aging, we found an expansion of

**Fig. 5 | FGF2 induces a reversible chondrocyte-like state in senescent hPnFbs in vitro. a** Representative image of sciatic nerves from *Sox10^CreERT2/+;R26^tdt/+* young and old mice. **b** Quantification of the number of SOX10 + /γH2AX + or VIM + / SOX10-/γH2AX + cells. *n* = 3 - 4, biological replicates. **c** Quantification of γH2AX + nuclei density in early and late passage hPnFbs. More senescent cells are detected in late passages. *n* = 6, biological replicates. **d** Quantification of EdU + nuclei density in early and late passage hPnFbs. More proliferative cells can be observed in early passages. *n* = 6 - 3, biological replicates. **e, f** Dot plots of senescence- or proliferation-related genes. **g** Representative images of senescent hPnFbs exposed to different FGF2 concentrations. Red arrows: FOXC2+ cells; green arrows: SOX9 + cells. **h** Quantification of EdU + nuclei density. *n* = 3 - 5 - 6 - 5, biological replicates. **i** Quantification of SOX9+ nuclei density. *n* = 6 - 7 - 7- 8, biological replicates. **j** Quantification of the density of SOX9 + /FOXC2 + nuclei. *n* = 6 - 7 - 7 - 8, biological

replicates. **k** Representative images of senescent hPnFbs exposed to different concentrations of FGF2. Green arrows point to γH2AX+ cells, red arrows point to FOXC2 + cells. **l** Quantification of γH2AX+ nuclei density. *n* = 6, biological replicates. **m** Representative images of senescent hPnFbs exposed to different FGF2 concentrations for 14 days followed by withdrawal for 7 days. Red arrows: FOXC2 + cells. **n** Quantification of SOX9 + /FOXC2 + nuclei density with (pink dots) and without (green dots) withdrawal for 7 days. *n* = 6 - 7 - 8 // 6 - 6 - 6, biological replicates. **o** Representative images of human peripheral nerve samples from young and older donors. Yellow arrows: SOX9+ cells; pink arrows: FGF2 + fluorescent signal; cyan arrows: SOX9 + /FGF2 + cells. **p–s** Quantification of the density of (**p**) FGF2 + cells, (**q**) SOX9 + cells, (**r**) FGF + /SOX9 + cells, and (**s**) nuclei in human peripheral nerves. *n* = 5, biological replicates. Data are presented as mean values +/− SD. Source data for panels is provided as a Source Data file.

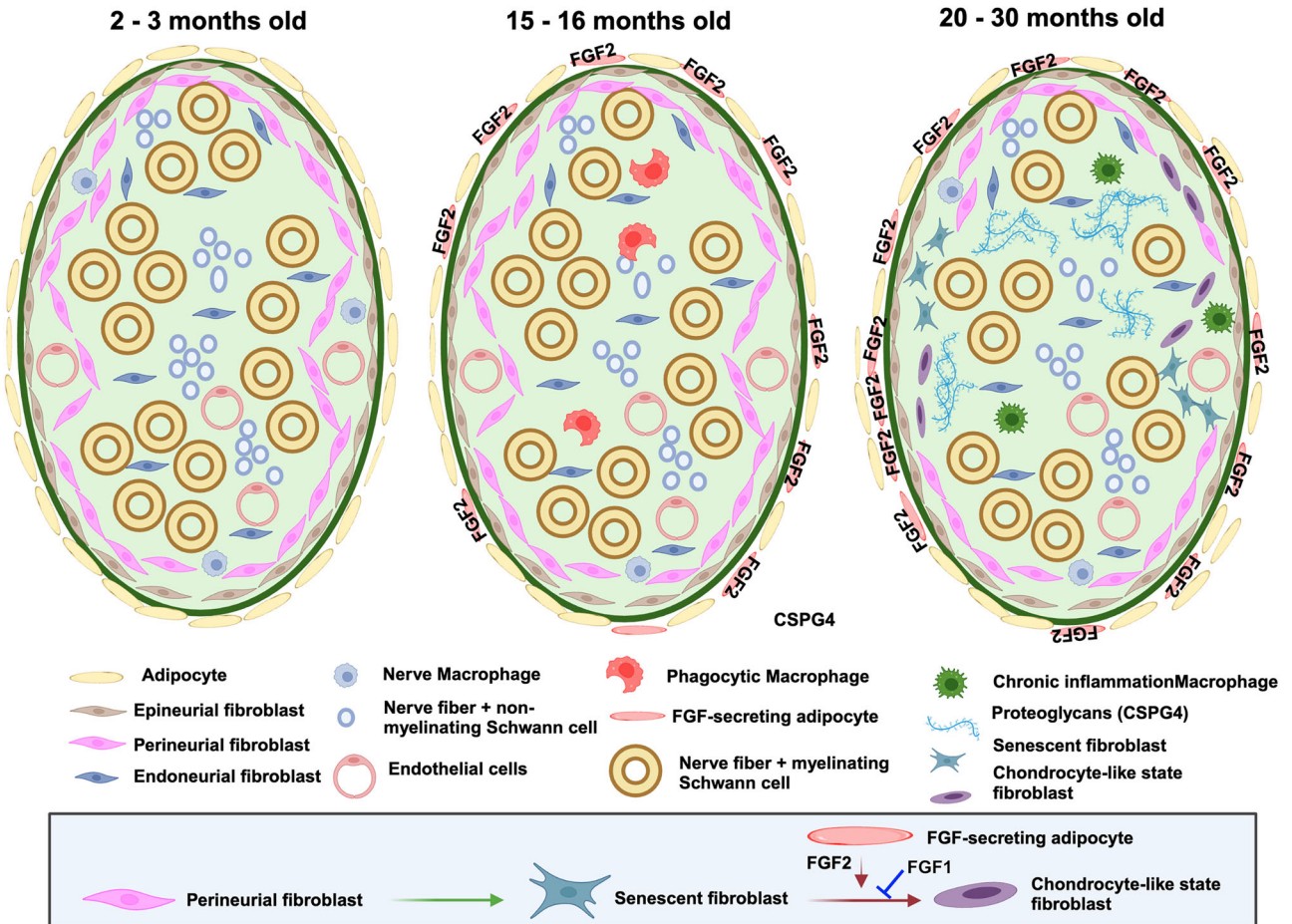

**Fig. 6 | Graphical summary of cellular changes during PN aging.** In sciatic nerves from 15-16 months old mice, increased number of macrophages with a phagocytic phenotype and FGF2 + adipocytes are observed. In sciatic nerves from 20-30 months old mice, increased macrophages with a chronic inflammation phenotype, FGF2 + adipocytes, senescent fibroblasts and Schwann cells, and chondrocyte-like state fibroblasts are observed. In the blue box a mechanism for activation of the chondrocyte like state, based in our in vitro studies, is shown. Perineurial fibroblast become senescent during aging, and increased FGF2 signaling activates the state. Created in BioRender. Preissl, S. (2025) https://BioRender.com/dwt19jy.

a fibroblast subpopulation that expressed a combination of chondrocyte-lineage markers (*Sox9, Foxc2,* and *Cspg4*). Reanalysis of a published dataset showed the presence of this subpopulation in nerves from P60 mice as part of the fibroblast cluster[24], however, the low abundance of this fibroblast subtype made them to pass underappreciated. In our case, this cluster was detected due to the increased presence of these fibroblasts in the sciatic nerves from aged mice. Interestingly, in tissue, we detected chondrocyte-like state fibroblasts, possibly derived from epi- or perineurial fibroblasts, in the endoneurium. This could point to increased fibroblast migration as another

feature of nerve aging. Independently of the nomenclature, these chondrocyte-like fibroblasts might be underlying the reported age-related increased deposition of proteoglycans like CSPG4[15], and or collagen[37,38], being possibly involved in the degeneration observed during aging of the PNS.

Besides degeneration and abnormal ECM deposition, aging is associated with the development of a global inflammatory state, also known as inflammaging[14]. In addition, previous studies have found increased Mφ numbers in the sciatic nerve of old mice[38,39], and proposed that a reduction in Mφ numbers also reduces nerve structural

**Table 2 | Age of the older animals included in this study**

| | |
|---|---|
| 30 months old | 3 |
| 28 months old | 1 |
| 25 months old | 2 |
| 22 months old | 1 |
| 20 months old | 1 |

and functional impairment associated with aging[40]. In the brain, M1-phenotype microglia has also been associated with neurodegenerative processes and aging[41]. In our work, we have described changes in Mφs populations during aging. In agreement with previous work on motoric nerves from young mice, we found resident steady state endoneurial and epineurial Mφ[29,30] representing the most prevalent populations in sciatic nerves from 2-3 months old mice. Interestingly, we confirmed Mφ activation during aging, but the prevailing state was different at the onset of aging (phagocytic), compared to advanced aging (chronic inflammatory). Abnormal myelin is known to activate a phagocytic phenotype in Mφs[40], and in both PNS and CNS, multiple myelin abnormalities are frequently observed. Therefore, it is possible that these phenomena are linked, with phagocytic Mφs trying to remove abnormal myelin and possibly, mSCs. While this possible acute recruitment of phagocytic Mφs happens at the beginning of the aging process, there seems to be a chronification of the age-related changes, that involves also a chronic-inflammatory profile of the nerve Mφs. Finally, it is important to remark that we observed a reduced number of phagocytic Mφs in the nerves from the oldest mice. This could be originated by an age-associated increase of Mφ malfunction, as it has been previously reported[42].

During aging, adipocytes and lipids accumulate in different organs[13], and cross-talk between adipocytes and peripheral glial cells have been previously reported[43]. In this work, we describe an increased number of adipocytes in the PNS of aged mice. One of the consequences of the age-related increase of adipocytes is an increased secretion of pro-inflammatory adipokines[44]. While FGF signals are not canonical adipokines, FGF2 has been considered a non-classical adipokine[45]. In our study, we found increased levels of FGF2 in sciatic nerve adipocytes, that, together with the increased number of adipocytes, leads to an increased secretion of this ligand. In addition, CSPG4 can bind directly to FGF2, increasing its signaling effect[46], and we have previously reported that CSPG4 expression increases with age in sciatic nerves from mice[15]. Altogether, these findings point to a dramatic increase in FGF2 signaling activity during aging of the PN.

FGF2 is known to induce chondrocyte differentiation in mesenchymal stem cells[19], but its effects on neural fibroblasts was unknown. Here, we confirmed the role of FGF2 as survival signal in healthy hPnFbs, where it reduces DNA damage[47]. In contrast, FGF2 effect on senescent hPnFbs seems to be quite different, activating the co-expression of chondrocytic markers like SOX9 and FOXC2. Notably, CSPG4 levels remained unchanged in both healthy and senescent hPnFbs upon FGF2 stimulation. While more exposure time to FGF2 might be required to increase CSPG4 expression, other signals might be involved in this mechanism. Interestingly, this expression change does not seem to lead to a full trans-differentiation process, but to a state because after FGF2 withdrawal the number of chondrocyte-like state fibroblasts is even below the baseline prior to FGF2 stimulation. This might be the reason why, in our human samples, we detected that SOX9+ cells also express FGF2, creating a positive FGF2 feedback loop to maintain the chondrocyte-like state. These results are quite encouraging as it seems that some of the age-related changes observed in peripheral nerves might be reversible.

In this regard, we found that FGF1 does not have the same effect as FGF2. Our finding highlights the relevance of re-evaluating the

knowledge acquired in young/healthy/homeostatic conditions, as FGF1 and FGF2 have been suggested to have similar roles in young healthy cells and tissues. Interestingly, high concentrations of FGF1 blocked the effect of FGF2. While further investigation is required to clarify the mechanisms, given that FGF1 and FGF2 share target receptors, competitive inhibition could be a possible explanation for our observations. With this information, FGF1 could be explored as anti-aging molecule in peripheral nerves, but what effects this molecule has on senescent hPnFbs and other PN cells, needs to be further explored.

Our data also indicates that aging-associated changes in the peripheral nerve can be observed as early as at 15-16 months of age. At this point, we detected reduced SC numbers and possibly nerve fiber density, together with increased myelin phagocytosis in Mφs, more FGF2+ adipocytes, and increased *Sox9* and *Foxc2* expression in fibroblasts. Conversely, CSPG4 deposition was only detected in the 20-30 months old age group. This points to CSPG4 deposition being a consequence of other initial aging alterations and makes futures studies of the effect of FGF2 on Schwann cells and nerve fibers, relevant. In any case, while further research will be necessary to disentangle the initial changes that trigger age-related impaired nerve function, we believe that axonal degeneration, together with myelinating Schwann cell loss, might initiate the aging cascade in peripheral nerves. It is possible that the loss of the neural compartment leaves a space that a combination of adipocytes and fibroblast try to "fill", further aggravating the age-related abnormalities.

Some limitations of the present study are that fluoromyelin and fluorescence microscopy might not be sensitive enough to identify all the age-related myelination changes, as most of them require electron microscopy to be detected. In addition, the number of mixed nerves from human samples is limited. Thus, we included pure sensory nerves in addition to mixed nerves in our analysis. Despite this limitation, we confirmed increased abundance of FGF2 + , FGF2 + /SOX9 + cells and decreased cell density in the human cohort.

In summary, we profiled cellular and molecular changes during PNS aging, using the mouse sciatic nerve as a model. In addition, we have characterized age-related changes in Mφ subtypes and showed an increase in the number of adipocytes in nerves from old mice. Finally, we propose that FGF2 signaling from these adipocytes activates PnFbs into a degenerative and, at least initially, reversible chondrocyte-like state and propose FGF1 as a possible antagonistic molecule to FGF2.

## Methods
### Experimental animals
All animal procedures were carried out in accordance with the guidelines stated in Directive 2010/63/EU of the European Parliament and of the Council of 22 September 2010 and were approved by the local authorities (Regierungspräsidium Freiburg; approval numbers X23-11R and G22-090). Experiments were performed on tissue obtained from young-adult (age: 2–3 months), adult (15-16 months), or old (age: 20–30 months) mice. *Sox10*-CreERT2 (*Sox10^CreERT2^*, Jax reference: 009074) mice were bred with *R26^tdt/tdt^* mice (Jax reference: 027651) mice to genetically trace Schwann cells. 3 mg Tamoxifen per kg of body weight was administered for 5 consecutive days, then tissue was harvested 9 days after the last injection. Mice were euthanized by cervical dislocation. Femoral sciatic nerves were sectioned from the sciatic notch to the trifurcation, by medial access through the biceps femoris in the posterior extremities. Nerves were washed and kept in phosphate-buffered saline (PBS) at 4 °C. The age of the older animals included in these studies can be seen in Table 2.

### In vitro studies on human perineurial fibroblasts
Primary human perineurial fibroblasts (hPnFbs) were purchased from ScienCell Research Laboratories (1710, Carlsbad, CA, USA), and growth in Geltrex (Thermo Fisher Scientific)-coated plates with Fibroblast

**Table 3 | Primary antibodies utilized in this study**

| Target Antigen | Host species | Provider |
|---|---|---|
| Cluster of differentiation 45 (CD45) 1:50 | Goat | AF114 R&D systems, Minneapolis, MN, USA |
| Cluster of differentiation 68 (CD68) 1:250 | Mouse | MCA1957 Biorad Hercules, CA, USA |
| Chondroitin sulfate proteoglycan 4 (CSPG4) 1:200 | Rabbit | ab129051 Abcam, Cambridge, UK |
| Fatty Acid Binding Protein 4 (FABP4) 1:100 | Rabbit | ab92501 Abcam, Cambridge, UK |
| Fibroblast growth factor 2 (FGF2) 1:100 | Mouse | MA5-15276 Invitrogen, Waltham, MA USA |
| Forkhead box protein C2 (FOXC2) 1:10 (Tissue), 1:30 (ICC) | Sheep | AF6989 Novus biologicals, Centennial, CO, USA |
| Gamma H2A histone family member X (γH2AX) 1:400 | Rabbit | 9718 Cell Signaling Technology, Danvers MA, USA |
| Glucose transporter 1/Solute carrier family 2 (GLUT1/SLC2A1) 1:200 | Rabbit | ab150299 Abcam, Cambridge, UK |
| Granulin (GRN) 1:100 | Rabbit | ab187070 Abcam, Cambridge, UK |
| Potassium Voltage-Gated Channel Subfamily C Member 2 (KCNC2) 1:100 | Rabbit | PA5-36072 Invitrogen, Waltham, MA, USA |
| Mannose Receptor C-Type 1 (MRC1) 1:100 | Rabbit | ab64693 Abcam, Cambridge, UK |
| SRY-Box transcription factor 10 (SOX10) 1:100 | Rabbit | ab180862 Abcam, Cambridge, UK |
| S100 calcium-binding protein B (S100B) 1:100 | Rabbit | ab52642 Abcam, Cambridge, UK |
| SRY-Box transcription factor 9 (SOX9) 1:250 | Rabbit | ab185966 Abcam, Cambridge, UK |
| SRY-Box transcription factor 9 (SOX9) 1:50 | Goat | AF3075 R&D systems, Minneapolis, USA |
| Transforming Growth Factor Beta Receptor 1 (TGBR1) 1:400 | Rabbit | 30117-1-AP Proteintech, Rosemont, IL, USA |
| Tyrosine Hydroxylase (TH) 1:100 | Rabbit | ab137869 Abcam, Cambridge, UK |
| Vimentin (VIMENTIN) 1:100 | Mouse | ab20346 Abcam, Cambridge, UK |

added at different concentrations every 24 h for 14 days. To explore the ability of hPnFbs to regress the chondrocyte-like state, FGF2 was added every 24 h for 14 days, and then normal fibroblast media with no FGF2 was added every 24 h for 7 more days.

## Immunofluorescence analyses
### Tissue immunofluorescence
**-Mouse nerves.** PFA-fixed nerves were cryoprotected in 15% sucrose in PBS overnight and then in 30% sucrose overnight. Next, nerves were embedded in Sakura Finetek Tissue-Tek optimal cutting temperature compound (OCT; Thermo Fisher Scientific, Walthamm, MA, USA), and plunge-frozen in liquid nitrogen. 5 µm sections were cut for immunofluorescence imaging (at least six sections per nerve were used for quantification). Slices were permeabilized with 0.01% or 0.1% Triton X-100 (Thermo Fisher Scientific) for 10 min at RT. Unspecific antibody binding was blocked with 6% donkey serum (Sigma-Aldrich, St Louis, MO, USA), and 1% bovine serum albumin (BSA; MilliporeSigma, Burlington, MA, USA) for 1 h at RT. Samples were incubated overnight with primary antibodies at 4 °C (see Table 3). After washing, samples were incubated for 1 h with Alexa-conjugated (488, 594, and 647) donkey secondary antibodies at 1:400 dilution at RT (Abcam, Cambridge, UK). Samples were mounted with VECTASHIELD Antifade Mounting Medium with DAPI (4′,6-diamidino-2-phenylindole, Vector Laboratories, Newark, CA, USA). For myelin fluorescent labeling, a green-fluorescently labeled fluoromyelin (F34651, Thermo Fisher Scientific) molecule was used at 1:300 concentration, for 20 minutes at RT, after the washing steps, post-secondary antibody incubation.

**-Human nerves.** Formaldehyde-fixed and paraffin-embedded (FFPE) human samples were obtained from surgical specimens in accordance with an Institutional Review Board–approved protocol (Ethic Commission of Albert-Ludwigs University of Freiburg: 1008/09). Informed consent was obtained from all patients, in accordance with the declaration of Helsinki. All ethical regulations relevant to human research participants were followed. All samples were evaluated to contain healthy peripheral motoric nerve tissue by a board-certified neuropathologist. The nerve type and age are the following:

The following FFPE human peripheral nerve samples were used for immunostaining: *Nervus accesorius* (17 years old), *nervus suralis* (24, 25, 31, 55, and 63 years old), *nervus femoralis* (33 years old), *nervus hypoglossus* (57 and 68 years old) and *nervus ulnaris* (65 years old). 5 µm sections were obtained with a microtome (Leica Biosystems). Paraffin sections were melted at 60 °C for 45 min, then deparaffinized and re-hydrated. Antigen retrieval was performed by boiling in 1× citrate buffer-based antigen retrieval solution (H-3300, Vector laboratories, USA) in a pressure cooker for 2 min. Blocking was performed using 6% Donkey serum (D9663, Sigma-Aldrich, USA) and 1% BSA (MilliporeSigma) in PBS at room temperature (RT) for 1 h. Primary antibodies were incubated at the concentrations specified in Table 3, overnight, at 4 °C. After washing, samples were incubated for 1 h with Alexa-conjugated (488, 594) donkey secondary antibodies at 1:400 dilution at RT (Abcam, Cambridge, UK). Samples were mounted with Fluoromount-G mounting media (Thermo Fisher Scientific).

**Immunocytochemistry.** Cells were initially washed in PBS, and when EdU was used, the Click-iT EdU reaction was initially perform as indicated by the manufacturer (C10350, Thermo Fisher Scientific). Afterwards, unspecific antibody binding was blocked with 1% BSA (MilliporeSigma, Burlington, MA, USA) for 30 minutes, at RT. Primary antibodies (see Table 3 for concentrations) were incubated afterwards in 1% BSA for 1 h at RT. After washing Alexa-conjugated (488, 594, and 647, Abcam, Cambridge, UK) donkey secondary antibodies were incubated at a 1:400 concentration in 1%BSA, for 1 h at RT. Finally, DAPI (Thermo Fisher Scientific) was added at a 1:5000 concentration for 15 minutes at RT.

Medium (2301, ScienCell Research Laboratories, ethical statement in https://sciencellonline.com/en/technical-support/ethical-statement/) in 5% $CO_2$. Passages 3 to 4 were considered as "early passage" based on higher proliferation rate and reduced DNA damage, while passage 8 and 9 were considered as "senescent" based on reduced to null proliferative activity and higher DNA damage. FGF2 (3718-FB-025, Biotechne, Minneapolis, MN, USA), FGF1 (AFL232-025, Biotechne) with heparin (PHR8927, Merck, Darmstadt, Germany), or a mix of all were

**Table 4 | Marker genes used to identify second level clusters of macrophages**

| Macrophage type | Marker genes |
| --- | --- |
| Endoneurial Mφ | *Lyve1-, Clec10a-, Retnla-, Mrc1 +, Csf1ra +, Cx3cr1 +, Tgfβr1-, Kynu-, Cd68-* |
| Chronic inflammation Mφ | *Tgfβr1 +, Kynu +, Cd68 +, Cx3cr1 +, Ccr2-, Cd74-, Hspa5-, Mrc1-, Csf1ra-, Cd44-, C1qa-, Grn-* |
| Circulating derived Mφ-2 | *Cx3cr1-, Ccr2 +, Cd74 +, Hspa5 +, Cd44 +, C1qa +, Grn +* |
| Circulating derived phagocytic Mφ | *Cd44 +, C1qa +, Grn +, Cd68 +, Cd74 +, Hspa5 +, Cxcr1-, Lyve1-, Clec10a-, Retnla-, Mrc1-, Csf1ra-* |
| Epineurial Mφ | *Lyve1 +, Clec10a +, Retnla +, Mrc1 +, Csf1ra +, Tgfβr1-, Kynu-, Cd68-* |
| Circulating derived Mφ-1 | *Cx3cr1-, Ccr2 +, Cd74 +, Hspa5 +, Cd44 +, C1qa-, Grn-* |

**Microscopy.** Fluorescence imaging was performed using a Fluorescence Microscope (Thunder Imager Leica Microsystems, Wetzlar, Germany) or a Confocal microscope (TCS SP8 X, Leica Microsystems). At least three images (at 20x magnification) from each section of each nerve or at least six images (at 20x magnification) from each well were analyzed for quantification.

### Image analysis

ImageJ was used to perform fluorescence quantification. Total cells were counted based on DAPI+ nuclei, which were counted was on size and level of circularity. Single and double positive cells were counted based on positive pixels for DAPI as well as for the fluorophore/s of interest. The density of extracellular proteins was estimated as the proportion of positive pixels against a zero-filled array (absolute map of positive pixels). Thresholds to eliminate background fluorescence were manually set and, along with imaging parameters, maintained for all data to allow comparison.

### Graphic representation and statistics

Prism 9.5.1 software (GraphPad, San Diego, CA, USA) was used for statistical analyses, preparation of graphs. Graphs show mean ± standard deviation. Statistical significance was determined using the Mann–Whitney $U$ or Kruskal-Wallis test (with posterior multiple comparison analysis). A $p$-value < 0.05 was deemed to indicate a statistically significant difference between medians or means, respectively.

### Nuclei isolation and single nucleus RNA-seq (snRNA-seq)

From each mouse, both snap frozen femoral sciatic nerves were thawed in nuclei isolation media (NIM, 250 mM Sucrose, 25 mM KCl, 5 mM MgCl₂, 10 mM Tris-HCl) with 25 mM DTT, 1X protease inhibitor (05056489001, Roche), 0.1% Triton X-100 and RNase inhibitor (N2515, Promega). Each pair of nerves was homogenized using a Kimble dounce tissue grinder (D8938, Merck) and the resulting suspension was filtered through a 30 μm CellTRics (Sysmex, MSA150914) into a LoBind Eppendorf tube (Eppendorf). Nuclei were pelleted (1000 x g, 10 min, 4 °C) and washed with NIM + DTT + Protease inhibitor (05056489001, Roche) and RNase inhibitor (N2515, Promega). After the wash, nuclei were resuspended in sorting buffer (1 mM EDTA, 0.2 U/μL RNase inhibitor (N2515, Promega), 2% fatty acid-free BSA (A7030-100g, Sigma) in PBS). Finally, the nuclei were filtered again through a 30 μm CellTrics (Sysmex, MSA150914), and DRAQ7 (#7406, Cell Signaling) was added to the nuclei suspension for nuclei detection by flow cytometry. 10,000-20,000 nuclei were sorted using an S3 Fluorescence-activated cell sorter (Bio-Rad). Nuclei were resuspended and used for snRNA-seq library generation following the Single Cell 3' Reagents Kits V3.1 User Guidelines (10x Genomics). Quality control and quantification of libraries was performed using a Tapestation (Agilent) and a Qubit fluorometer (Life Technologies). Libraries were sequenced using a NovaSeq 6000 sequencer (Illumina).

### snRNAseq data processing and analysis

Sequencing data were preprocessed using 10x Genomics Cell Ranger-7.0.0[48]. FASTQs were aligned to the mm10 reference genome, including reads for intronic and exonic regions (10x Genomics). Nine snRNA-seq datasets from sciatic nerves at three time points measured in triplicate (3, 16, and 24 months old mice) were individually preprocessed and loaded into Seurat[49] (v4.3.0.1). All datasets were downsampled to 32k UMIs per nucleus. Nuclei with 800 to 7000 detected genes and less than 5% mitochondrial RNA content were kept for downstream analysis. Each dataset was then normalized and scaled, using 6200 variable features as selected by a variance-stabilization transformation (VST). DoubletFinder[50] (v2.0.3) was applied using the first 30 principle components (PCs), and using an estimated number of doublets based on a Poisson distributed doublet formation rate of 7%. (pN = 0.25, pK = 0.05). Harmony[51] was used for integration.

**Giving identity to clusters.** 26 of the initial Leiden clusters were annotated based on marker gene expression of each cluster (Table 1), and 5 doublet clusters were removed based on shared expression of robust marker genes from multiple cell types, and 2 residual skeletal muscle clusters were removed as not part of the sciatic nerve. Markers for Perineurial and Endoneurial Fibroblasts, Adipocytes, Mesenchymal Stem Cells (mSC), Chondrocytes, Macrophages, Epineurial cells, B-cells, Pericytes, and others were co-opted to delineate expression modes[24,25] (Table 4). To reduce unwanted variation in the analysis, primarily from cell doublets, sub-clustering and re-clustering was performed to better refine the expression of these cell types.

**Cluster refinement of snRNA-seq data.** Re-clustering was performed on the clusters listed above using the same initial pre-processing steps outlined above. Briefly, after sub-selecting for clusters of interest, the subset data was re-normalized and scaled using a variable feature count of 3,100 genes and re-clustered using a Leiden resolution of 0.3. The datasets were then reintegrated by time point using Harmony[51]. Unsupervised marker gene detection was then employed via the 'FindAllMarkers' function, using a log fold change threshold of 0.25 on genes that are detected in at least a quarter of each population during the pairwise cluster comparisons. These clusters are then re-labeled, disregarding any detected doublet clusters.

**Cell-Cell interaction pathway analysis.** To measure cell-to-cell interaction pathways, Cellchat[32] (v1.6.1) was applied. Overexpressed signaling genes and interactions associated with each cell group were detected, using the in-built mouse database. Communication probabilities between interacting groups were computed using default parameters, filtering for groups with at least 10 participating cells in each cell group. The probabilities in signaling pathways were similarly computed and summarized on all related ligands and receptors, using defaults. By summarizing the probabilities for each communicating pair of cell groups, the aggregate network is computed, returning summarized interaction weights between any two cell groups. Interaction plots were then generated for each detected pathway, as well as comparative heatmaps depicting the number of differential interactions in each signaling pathway.

**Pseudotime analysis of Chondrocyte-state fibroblast origin.** A pseudotime analysis was performed on the chondrocytes using Monocle[52] (v1.3.1), with initial filtering of cells with low gene count.

Cells were clustered using the Louvain algorithm on a k = 300 nearest neighbor graph, with a q-value cutoff of 1. The cell trajectories were then learned, disregarding any hard partitioning from the clustering to give a seamless single trajectory graph. By using the perineurial, epineurial, and endoneurial fibroblast clusters as the root cells, a cell ordering using the UMAP embedding was performed, with chondrocyte-like state fibroblasts as the final cells.

## Reporting summary

Further information on research design is available in the Nature Portfolio Reporting Summary linked to this article.

## Data availability

To facilitate the visualization and exploration of the single-nucleus RNA-seq data, we developed a docker (docker.com) container to run a CELLxGENE instance[53]. This ensures reproducibility and accessibility. This container includes the complete snRNA-seq dataset and the second-level Adipocyte and Macrophage clustering. Instructions on how to use the container can be found at (https://github.com/ usegalaxy-eu/docker-cellxgene-mouse-sciatic-nerve). To enhance accessibility, this container was integrated to the European Galaxy server (usegalaxy.eu) as an interactive tool[54]. This tool launches CELLxGENE with the selected dataset and provides a web-based interface for visualization. This tool and its instructions can be accessed via (https://usegalaxy.eu/?tool_id=interactive_tool_cellxgene_ mouse_sciatic_nerve&version=latest). Raw sequencing data is deposited to GEO under accession number GSE280857. Processed data including UMAPs from first and second level clustering are deposited on Zenodo (https://doi.org/10.5281/zenodo.14900774). Source data for panels is provided as a Source Data file. Source data are provided in this paper.

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

## Acknowledgements

We thank, Dr. Achim Lother for his scientific advice. We thank the Max Planck Institute of Immunobiology and Epigenetics Deep Sequencing Facility (Freiburg, Germany) for providing sequencing services. We acknowledge the support of the Freiburg Galaxy Team. We acknowledge the microscopy facility SCI-MED (Super-Resolution Confocal/ Multiphoton Imaging for Multiparametric Experimental Designs) at IEKM, Freiburg, for providing expertise and access to imaging setups and analysis workstations. S.P. was supported by the DFG through project B03 of TRR359 (project ID 491676693), project S03 of CRC1425 (project ID 422681845) and project A07 of GRK2344 (project ID 322977937). S.J.A. is funded by the DFG through the Heisenberg Program (AR 732/3-1), project grant (AR 732/2-1), project A08 of CRC 992 (project ID 192904750), and Germany's Excellence Strategy (CIBSS – EXC-2189 – project ID 390939984). F.S.W. was supported by the DFG via project P13 of CRC1425 (project ID 422681845), P7 of FOR5807 (project ID 537609931), and an Emmy-Noether fellowship (project ID 412853334). L.M. was supported by the Berta-Ottenstein-Program for Clinician Scientists, Faculty of Medicine, University of Freiburg.

## Author contributions

Study design was done by S.P. and LH. Experimental procedures were executed by D.S., E.S., S.H., C.D., M.H., S.P.F. and L.M., and LH. Data acquisition, analysis, and interpretation were carried out by D.S., M.T., A.N.N., B.G., S.P., and L.H. Figures and manuscript were prepared by S.P. and L.H. The Manuscript was prepared and discussed by F.S.W., S.J.A., M.P., S.P. and L.H. The final version of the manuscript was approved by all authors.

## Funding

## Competing interests

The authors declare no competing interests.
