## [Transparent Peer Review file · Nature Communications]

FGF signaling induces a chondrocyte-like state of peripheral nerve fibroblast during aging

Corresponding Author: Dr Luis Hortells

Version 0:

Reviewer comments:

Reviewer #1

(Remarks to the Author)

The aim of the study was to characterize non-neural cells, like nerve macrophages, fibroblast-like cells and adipose cells during aging and degeneration of mouse peripheral nerves. The authors performed single nuclei RNAseq, aiming to identify distinct signature genes expressed in 18 clusters at distinct time points (2-3; 15-16 and 20-30 months of age). Focusing on the identified distinct subtypes of macrophages, fibroblasts and adipose cells, the authors tried to partially validate data by immunohistochemistry and, finally, generated a model, how the different populations change and interact during aging. For instance, the authors describe a stepwise change of macrophages activation during aging starting with an "M2"- phenotype that adopts a transient phagocytic signature and, finally, showing an "M1" – like state in aged nerves. Furthermore, fibroblasts adopt a senescent feature, eventually developing into a chondrocyte-like state. Based on their ligand-receptor analysis, the authors speculate that possibly adipose cell-borne FGF2 may drive senescent nerve fibroblasts to adopt this chondrocyte-like state. In an in vitro approach designed to mimic the conditions of aging nerves, human senescent fibroblasts were induced to adopt a chondrocyte-like phenotype following stimulation with FGF2. This effect, however, could be blocked by administering high doses of FGF1. The authors proposed this mechanism as a possible translational strategy to mitigate nerve aging.

Generally, the topic is of high interest, particularly for communities with increasing life expectancy, as degenerative processes in aged peripheral nerves may lead to reduced mobility and poor life quality. Furthermore, due to increased care dependency socio-economic problems emerge.

A) General major criticism on the concept of the study:

The authors here present molecular changes that occur in distinct, non-neural cell types of peripheral nerves during young, adult and older ages. Although these data may be quite interesting on a purely descriptive level, there is no attempt to prove whether these changes are causally linked to the well known (but here not described) degenerative changes typical for aging nerves. Consequently, the data presented do not allow to discriminate whether the molecular changes are cause or consequence of degenerative processes within the aging nerve. For instance, it is conceivable that degenerative features could hypothetically be initiated by senescent neurons and axons which could – like a Wallerian-like process - lead to the here observed changes, thus being rather the consequence than the primary cause for degeneration. While it is, nevertheless, possible that nerve degeneration may be forced by the molecular changes described here, there is – as said – no attempt to decipher such a secondary amplification mechanism, for instance by targeting the identified molecules and/or the respective cell types or the proposed cell-cell interactions. This is a serious drawback of the study that substantially limits its contribution to understand peripheral nerve degeneration at advanced ages.

B) Major criticism on experimental performance and interpretation:

a) the presentation of the snRNA-seq data does not follow the state-of-the-art frame: additional genes for the different cellular

populations are missing, as well as heat map analysis, differential gene expression analysis, ViolinPlots and FeaturePlots. Based on these omissions, the study is far away from being a “sciatic nerve aging atlas” as the authors claim (see for comparison, Gerber et al., 2021; Yim et al., 2022).

b) established macrophage markers like *Csf1r* and *Adgre1* are not represented in the ViolinPlots in Fig. 1. CD45 is not a specific marker for macrophages, but a global immune marker.

c) LYVE1 is also associated with epi-/perineurial macrophages and not confined to perivascular ones (see Ydens et al., 2020)

d) the GLUT1+ immunoreactive, more linear profiles in Fig. 2E (middle; “16 m.o.”) do not look like macrophages at all. Rather, they look like blood vessels being in line with the established finding that GLUT1 is a marker for endothelial cells (see also Gerber et al., 2021). In the left and right panel of Fig. 2E, the positive structures look like slim perineurial cells, without any hint for phagocytosis, while phagocytic cells are usually more rounded and not arranged as linear structures. Thus, designating GLUT1 as phagocytic marker does not at all fit to the immunocytochemical staining, corroborating the overall questionable validation approaches.

e) the M1/M2 nomenclature is old-fashioned and does not represent the recent state of the art. It was previously used at “pre-single cell/nucleus-analysis” times as an approach to try to classify “good” and “bad” macrophages but is outdated at times when more sophisticated methods (like snRNA-seq) are available.

f) *Cspg4+* fibroblast have been found in datasets from other laboratories. However, they are described already in 60d-old mice (<https://snat.ethz.ch/search.html?q=cspg4>; Gerber et al., 2021).

g) there is a plethora of literature that describes the pathological and pathophysiological features of aging nerves: Ceballos et al., 1999; Cowen et al., 2005; Leblhuber et al., 2011; Anish et al., 2015; Ward et al., 2015, 2016; Canta et al., 2016; Moldovan et al., 2016. Unfortunately, none of the papers indicated is cited, although it would help the reader to understand the aim of the study and would represent a comprehensive understanding of the topic by the authors.

h) According to the authors view, FGF2 expression appears to be a pivotal player regarding the development of chondrocyte-like cells, indicative of aging fibroblasts. It is, therefore, essential to verify the expression by qPCR and/or Western blot analysis in a time-dependent manner.

i) A range of 20–30 months is quite broad. Could the authors clarify the specific ages and the rationale of selection of the ages investigated in the study?

C) Minor, but significant problems:

a) When addressing nerve adipocytes, the study by Sundaram et al. (2023) should be considered. It is very well known that adipocyte-like cells are at the edge of the nerve (peri-/epineurium), while it appears that the authors are surprised by that, as they claim: “Remarkably, these adipocytes where (should read: were) detected mainly in the epi- and perineurium, making local adipocyte-hPnFbs interaction feasible (Fig 4L). Importantly, fibroblasts can also be – as opposed to adipocytes – endoneurial.

b) Introduction: what are “circulating phagocytic macrophages”?

c) Fig. 6: the schematic drawing of the nerve does not fit with the nerve architecture in mice. Rather, the subdivision into individual fascicles is typical for human nerves; maybe a scheme of a human nerve may have been erroneously taken as a base.

Furthermore, in the schematic summary it looks like as FGF2 is derived from senescent fibroblasts.

d) Fig. 1a: the nerve isolation scheme does not represent the position of the sciatic nerve. It is a ventral view and would allow access to the femoral rather than the sciatic nerve.

e) How do the authors conclude that *Tgfb1* (Transforming growth factor beta receptor 1) is a specific marker for macrophage activation in the PNS? Is there specific literature available? Moreover, in Table 1, the authors erroneously name Tgf... as “tissue growth factor...”.

Reviewer #2

(Remarks to the Author)

Reviewer #3

(Remarks to the Author)

In their manuscript the authors performed a single cell characterization of rodent large peripheral nerves at ageing and identified that perineurial and epineurial fibroblasts exhibited a chondrocyte-like transcriptional profile. Using a combination of immunohistochemistry and in vitro studies, the authors identified FGF2 produced by adipocytes increasingly with age as a likely candidate inducing this chondrocyte-like transcriptional profile. Overall, this is a well-written manuscript reporting an interesting and technically well-founded observation potentially contributing to age-related changes in peripheral nerves. However, the functional relevance of the observations remains unknown and the overall impact of the manuscript therefore appears very limited.

Major points:

1. The manuscript currently reports an interesting finding but remains fully descriptive although many straight-forward follow-up experiments could be envisioned. What exactly is the functional relevance of the proposed chondrocyte-like transcriptional state induced by FGF2? Does this directly or indirectly affect the function of the peripheral nerves? For example, does this impair nerve conduction or propagate axonal loss or myelin loss in ageing? Would FGF2 overexpression or addition induce similar changes and then functional impairments without ageing. Or would such chondrocyte-like cellular states alter the stiffness / flexibility of the nerves? Or would FGF2 activation induce endoneurial or perineurial fibrosis or swelling or cellular proliferation? As it stands, Fig 3H shows relative proportions but not change in cell numbers or nerve area etc.
2. And if the answer to any of the prior questions is yes, would deficiency of FGF2 or of its down-stream signaling pathways in nerve fibroblasts then abrogate age-induced pathology? An FGFR-conditional mutant is available.
3. An alternative or additional way of increasing the relevance of the finding could be to aim for human confirmation. Observing similar alterations in peripheral nerves (e.g. sural nerve biopsies) of human individuals with polyneuropathies vs. 'non-diseased' ageing vs. young individuals would support the relevance of the findings and improve credibility.
4. With their simplified M1 vs M2 macrophage phenotyping, the authors omit important parts of the literature demonstrating the existence of endo- vs. epineurial macrophages with distinct transcriptional phenotype and distinct ontogenetic origin (DOIs: 10.1038/s41467-020-16355-w, 10.1038/s41593-020-0618-6, 10.1073/pnas.1912139117). The authors should also cite other papers than Yim et al 2022 Nat Neurosci reporting scRNA-seq characterization of peripheral nerves (e.g. DOI 10.7554/eLife.58591)

Minor points:

1. Labels and font sizes in Fig 2A-D should be bigger.
2. Fig 2F is a sub-optimal way of depicting this. Consider depicting proportions of total cells (e.g. in stacked barplots) rather than individual cells.

Reviewer #4

(Remarks to the Author)

The article addresses a critical question in the aging of the Peripheral Nervous System (PNS): what cellular mechanisms drive nerve degeneration and inflammation, and what changes occur in the PNS extracellular matrix (ECM)? These are highly relevant questions for both the fields of peripheral nerve biology and aging. The authors present a cellular atlas of PNS aging and identify fibroblasts, macrophages, and adipocytes as the most affected cell populations. Furthermore, they characterize distinct macrophage phenotypes in the sciatic nerve during aging and confirm the presence of "inflammaging" in the aging nerve, a phenomenon already described in the literature.

A particularly interesting aspect of the manuscript is the investigation into the transformation of fibroblasts into chondrocyte-like cells during aging. The proposed link between pro-inflammatory adipocytes, FGF signaling, and the activation of chondrocyte-like fibroblasts is compelling. The manuscript suggests that Schwann cell dysfunction and axonal degeneration may result from ECM changes driven by altered adipocyte signaling. The authors' focus on epineurial adipocytes and their impact on fibroblasts and ECM remodeling is a unique and underexplored angle in the field of aging in the peripheral nerve. In principle the manuscript is suitable for publication, but there are several areas that require improvement:

1. Introduction: The description of SOX9 as a "marker" for endoneurial fibroblasts and satellite cells during repair could be clarified. The term "marker" implies specificity, which may not be accurate. A more precise phrase, such as "SOX9 is expressed in these cell types under specific conditions," would be preferable.
2. Figure 1D: The data suggest a reduced presence of B cells in aged nerves, but additional evidence is needed. An extra panel in Fig. S1 showing the nuclei density of the B-cell population could clarify this. Additionally, a shift in the EC1 population at 16 months is noted but not discussed. It would be useful to include a brief comment on this observation, as the shift in macrophage populations is already addressed.
3. Inflammaging: While inflammaging and macrophage activation are discussed, the addition of a reference such as Büttner et al. (2018), which describes an increase in macrophages in aging nerves, would strengthen the argument.
4. Macrophage Phagocytosis: GLUT1 is used as a marker for phagocytic activity in macrophages, with increased phagocytosis noted during aging. It would be valuable to test whether macrophages are phagocytosing myelin, perhaps using F4/80 and Fluoromyelin co-staining in sciatic nerve sections. Since abnormal myelin is mentioned in the discussion, additional experimental evidence here would be beneficial.
5. Figure 2E: Phagocytosis is increased in middle-aged animals but not in old animals. This discrepancy should be

discussed. One possible explanation is that macrophage dysfunction may occur with aging, preventing effective phagocytosis. This hypothesis should be explored further.

6. Fibrosis and Calcification: The development of fibrosis and calcification in organs such as the heart and lung is well-documented during aging. A reference supporting this in the context of peripheral nerve aging would be useful.

7. Collagen Expression: The increased expression of Col1a1 and Col2a1 in the aging nerve could be validated through immunofluorescence staining to further support the idea of ECM remodeling. This would provide additional insight into the spatial distribution of these collagens during aging.

8. Figure 2: The concentration-dependent increase in the chondrocyte-like fibroblast state is an important finding and should be explicitly mentioned in the text.

9. FGF2 Expression: The expression of Fgf2 could be further contextualized by adding fold-change or ratio data for the three timepoints, which would strengthen the physiological relevance of the findings and justify the use of high concentrations of Fgf2 in the cell culture model.

10. Graphical Abstract: The reversibility of the chondrocyte-like state, demonstrated by the ability of Fgf1 (at sufficient concentrations) to revert the transformation, should be indicated in the graphical abstract. Adding a dashed line or other visual cue in the bottom panel could highlight this potential therapeutic avenue.

11. Methods Section: The number of animals used to isolate enough nuclei for sequencing is not specified. This should be included for transparency.

12. Aging and Early Changes: The data suggest that age-related changes in the PNS begin as early as 15-16 months. The discussion could benefit from further exploration of what drives these early changes and their implications for later-stage aging.

Figure Labeling and Presentation:

- Color Choices: Red and green color schemes (e.g., in Figure 2B/C) may not be optimal due to red-green color blindness. Alternative colors should be considered.

- Consistency in Age Group Labeling: The age groups are mostly labeled as 15-16 months, but Figure 2E only uses "16 months." Consistent labeling is important for clarity.

- Typographical Issues: In Figure 2A, the label for "inflammation" contains a typo, and the label in 2F should have the timepoint reversed. Additionally, the font size in panels 2A-D and 2F should be increased for readability.

- Missing Letters in Figure 2D Labels: Some letters are missing from the labels in panel 2D, which should be corrected.

Conclusion:

The manuscript presents a valuable contribution to understanding the cellular mechanisms underlying PNS aging, with a novel focus on ECM remodeling and fibroblast transformation. However, some additional data, clarifications, and improvements in figure labeling are necessary to strengthen the overall impact of the study.

Version 1:

Reviewer comments:

Reviewer #2

(Remarks to the Author)

While some of the referees' concerns have been addressed by the authors, and the efforts intended to improve the manuscript are acknowledged, several specific points remain that require further clarification or experimental support:

a) Despite some attempts in the revised manuscript, the data still does not allow for a clear distinction between whether the molecular changes are a cause or a consequence of degenerative processes within the aging nerve. Further clarification or additional evidence would be necessary to address this issue. For example, if histopathological changes in the peripheral nerve occur earlier (see also comment b), but CSPG4 deposition and FGF expression are observed at later stages, could these be rather considered as a consequence of degeneration? Additionally, the statement that "both SOX9+ cell density and CSPG4 area were significantly increased starting from the 15-month time point (Fig. 3F and F)" is confusing, as the quantification does not support this. The significance was only observed at the 20-24 month time point, and this discrepancy should be addressed.

b) The new S100B staining presented in the manuscript (Figure S1A) does not appear convincing compared to existing literature (e.g., DOI: 10.1371/journal.pone.0123278, DOI: 10.3390/ijms18020263). It may be beneficial to include additional markers, such as Sox10, as previously performed by the authors, to better quantify Schwann cell numbers. Furthermore, quantifying the relative numbers of S100B+ cells per total DAPI+ nuclei does not fully capture a loss of Schwann cells, as it may be influenced by an increase in the number of other endoneurial cells (such as macrophages and fibroblasts) in aging nerves. The authors also state that two defined hallmarks of peripheral nerve aging are the loss of nerve fibers and the loss of Schwann cells. However, other studies have demonstrated additional pathological changes associated with aging, including demyelination, Schwann cell proliferation and onion bulb formation. These features are also important hallmarks of age-related peripheral nerve pathology and should be acknowledged to provide a more comprehensive view. Therefore, the conclusion that "aging changes in the peripheral nerve start as early as at 15–16 months of age" requires further validation. Supporting evidence such as neurofilament (NF) and P0/MBP staining on cross-sections (see also DOI: 10.12659/MSM.918277) or ultrastructural analyses using high-resolution electron microscopy (EM) would strengthen the hypothesis.

c) The authors present new data on FGF expression counts per animal based on the snRNA-seq approach. However, as

FGF2 expression appears to be the pivotal player in the development of chondrocyte-like cells, it is essential to verify the expression by another approach (qPCR and/or Western blot analysis) in a time-dependent manner.

d) The immunohistochemical expression of SOX9 and CSPG4 should be validated with a pan-fibroblast marker to confirm the fibroblast identity of the proposed chondrocyte-like cells (Figure 3). Furthermore, if the authors suggest that the most probable origin of these chondrocyte-like fibroblasts is the perineurial fibroblast (PnFb) population, and that endoneurial fibroblasts are the least likely source, this raises an important question: how is the age-dependent signaling mediated within the endoneurium - the primary site of nerve damage? Do the authors propose that fibroblasts migrate from the perineurium to the endoneurium? At present, the manuscript does not really provide direct evidence supporting that the chondrocyte-like fibroblast state most likely originates from PnFbs in the sciatic nerve of aged mice. Clarification and additional validation would be important to support this conclusion.

d) In Figure 3H, the SOX9/KCN2 immunoreactive profiles (fibroblasts) are indicated by red arrows. However, these profiles are detectable in the endoneurium, rather than the epineurial layer as indicated in the quantification. This discrepancy should be clarified to ensure accurate interpretation of the data.

f) Regarding the human data presented in Figure 5I: could the authors clarify whether the observations are based on comparable peripheral nerves, and whether differences in nerve type may influence their findings? Why are less nuclei present in the nerve biopsy (Human -Old)? To more accurately confirm and validate the specific cell populations that express FGF2 and SOX9, co-staining with established fibroblast and adipocyte markers would strengthen the conclusions. The presence of FGF2-expressing adipocytes within the nerve endoneurium is unusual (Figure 5I). Could the authors provide an explanation or relevant literature to support this observation? Furthermore, in the mouse dataset, SOX9+ fibroblasts and FGF2+ adipocytes are shown as two independent populations. Why are these markers quantified together in the human samples (SOX9+ / FGF2+ cells; Figure 5I'')?

g) Regarding the validation of phagocytosing macrophages using GLUT1 immunohistochemistry: as mentioned in the previous review, the observed profiles resemble blood vessels more than phagocytosing macrophages. GLUT1 is known to be expressed in endothelial cells (see Gerber et al., 2021), and there does not appear to be a complete overlap between CD45 and GLUT1 staining. Additionally, similar GLUT1-immunoreactive profiles in Figure 3H are interpreted as perineurial fibroblasts. Could the authors clarify the rationale for using GLUT1, especially given that this marker is neither present nor indicated in the phagocytic macrophage cluster (defined by Cd44, C1qa and Grn)? Based on this, GLUT1 does not appear to be a reliable marker for identifying phagocytosing macrophages and its use in this context should be reconsidered.

h) Several of the nerve sections lack optimal quality, as samples are not perfectly longitudinally oriented (e.g., Figure 2F), and some display structural disruption (e.g., Figure S3H). This is not state-of-the-art and complicates interpretation. Improving section quality would enhance the reliability of the histological analysis.

Minor:

a) Wording: "We detected higher numbers of macrophages with intracellular myelin was found in sciatic nerves from...". (was found should be omitted)

b) While the authors mention moving away from the M1/M2 nomenclature, several parts of the manuscript (e.g., Figure S3, Fig 6) still contain the M1/M2 terminology. For consistency, revising the manuscript accordingly is recommended.

c) It would be helpful to include the relative percentages of fibroblast subtypes across ages (similar to how macrophage subclusters are presented in Figure 2).

d) Could the authors clarify why tyrosine hydroxylase (TH), which is particularly expressed in peripheral sympathetic neurons, was used as a marker for PNS axons in Figure S1? It would be helpful to understand the rationale for choosing TH over more commonly used axonal markers in the sciatic nerve, such as neurofilament.

e) For clarity and transparency, the age range (specific ages and rationale) of the animals used in this study (20-30 m.o.) should be explicitly stated in the Materials and Methods section.

f) The authors state that "...we detected a previously undescribed population of fibroblasts that expressed a combination of chondrocyte-lineage markers (Sox9, Foxc2, and Cspg4)...". However, as noted in the previous review, the expression of these genes is detectable in 60-day-old mice (see <https://snat.ethz.ch/index.html>). It would be helpful to clarify how this population is considered novel in light of existing data.

g) The authors mention that the "chronic inflammation macrophage" cluster (Tgfr1+) aligns with the findings of Zhigang Lei et al., 2024 (JCI Insight). However, this referenced paper primarily describes macrophages in the context of cancer or chronic infection, not in PNS tissue. Is there other literature available demonstrating Tgfr1 expression as a marker for macrophage activation in the PNS? Additionally, this referenced paper focuses on PD-1 expression on macrophages in "chronic inflammatory tissues", rather than specifically addressing "chronic inflammatory macrophages". Lastly, are similar markers expressed in the Tgfr1+ macrophage cluster in the PNS compared to the cancer macrophage subtype described in the referenced paper? For instance, does this cluster show high expression of Pdcd1 or other related markers?

h) Figure S2H (middle panel): does the strongly reduced Fluoromyelin intensity indicate overall myelin loss/demyelination in

16 m.o. sciatic nerves or is this signal loss due to technical reasons?

Reviewer #3

(Remarks to the Author)

My previous concern with the manuscript was the descriptiveness of the data since no functional relevance of the FGF2 signalling pathway was provided. The authors do not provide any additional data in this regard.

Instead, the authors now provide FGF2 / SOX9 staining of human peripheral nerve specimen from 2 young and 3 aged individuals and depict representative stainings confirming their key observation in humans in vivo. This does improve the credibility of the finding. However, the human data remain limited in several aspects:

- a) In the online methods text, I do not find information on how the proportion of FGF2/SOX9 stained cells was quantified. Is this stained cells per all DAPI+ cells?
- b) If this (apparently) is cell density, how would absolute number look?
- c) I consider a 2 vs 3 comparison without testing for significance inadequate.
- d) It remains debatable how well mixed nerves (e.g. ulnar nerve) can be compared with purely motor nerves (e.g. hypoglossus, accessory nerve) as outlined in the methods; especially given that the Schwann cell phenotype (and myelin protein composition) differs between motor and sensory nerves / fibres (e.g. Yim 2022 Fig 5)

In essence, in my opinion the manuscript has improved but with further need for improvement

Reviewer #4

(Remarks to the Author)

I have read the authors' rebuttal and am satisfied that they have adequately addressed all reviewer comments. At this stage, I do not find it necessary to request additional experiments. Overall, I am positive about this manuscript and believe it is, in principle, suitable for publication.

Version 2:

Reviewer comments:

Reviewer #2

(Remarks to the Author)

Although the authors have addressed most of the referees' concerns and their efforts to enhance the quality of the manuscript are acknowledged, several specific issues related to data validation remain:

- the referee is still surprised that the authors use GLUT1 as a marker for phagocytosing macrophages. Slc2a1 (Glut1) is not detectable in the macrophage clusters (see Violin plots in Figure 1e) but is present in endothelial cells (EC) and pericytes. This may also account for the blood vessel-like morphology observed in the immunohistochemical stainings (see previous comments to the authors). It would be helpful to display the separated staining channels in Figure S3H to clarify whether macrophages are simply localized near blood vessels, as frequently observed in high-resolution electron microscopy studies and as suggested by the current staining, or really express Glut1. Co-staining with an additional blood vessel marker would also be advantageous.

- as previously noted, to confirm and validate the specific cell populations expressing FGF2 and SOX9, co-staining with well-established fibroblast markers would strengthen the conclusions. Pdgfra or Cd34 are recommended markers for fibroblasts.

- The previous comment to the authors stated: "In Figure 3H, the SOX9/KCN2 immunoreactive profiles (fibroblasts) are indicated by red arrows. However, these profiles are located in the endoneurium rather than in the epineurial layer, as indicated in the quantification. This discrepancy should be clarified to ensure accurate interpretation of the data." The authors responded: "As mentioned before, migration is indeed an interesting possibility and we have included it now in the discussion." Unfortunately, this response does not address the concern regarding the accurate interpretation of the data. To distinguish between epi-, peri-, and endoneurial cells, different arrowhead colors should be used in the representative staining, and the SOX9/KCN2 immunoreactive profiles within the endoneurium should also be quantified and presented.

Reviewer #3

(Remarks to the Author)

The purely technical aspects of my previous concern were addressed and the description of the FGF/SOX9 quantification has improved.

Unfortunately, however, the limited number of human samples remains unaddressed. My previous wording here was: 'I consider a 2 vs 3 comparison without testing for significance inadequate'

The authors' attempt at discussing this remains disappointing since my concern would have been straightforward to address. Although sural nerve biopsies are performed rarely, every neuropathology department will hold sample banks of human sural nerve biopsy sections across a wide age range.

Increasing the number of stained samples in Fig 5I would thus have been easy and I am still convinced that this would improve the manuscript

Version 3:

Reviewer comments:

Reviewer #2

(Remarks to the Author)

The aspects of my previous concern have now been addressed in the revised manuscript.

Reviewer #3

(Remarks to the Author)

I find that the authors have adequately addressed my concerns regarding the human nerve-based data and their statistical quantification

I endorse publication

08.04.2025

MS# NCOMMS-24-68816

Dear colleagues,

We would like to thank you for the constructive comments and consideration of our manuscript “FGF signaling induces a chondrocyte-like state of peripheral nerve fibroblast during aging”. For this resubmission, we performed additional analysis of the snRNAseq analysis, and setup an interactive online portal for easy data exploration. We have also performed further experiments, revisited the bibliography, and modified the text in response to your comments and suggestions.

Specific comments from each reviewer have been addressed and described below. Specific changes/additions to the manuscript have been highlighted (in yellow) in the markup version of the text.

We believe that the revised manuscript addresses the reviewers' comments, which have helped to improve the paper.

Thank you.

Response to reviewer #1

A) General major criticism on the concept of the study:

The authors here present molecular changes that occur in distinct, non-neural cell types of peripheral nerves during young, adult and older ages. Although these data may be quite interesting on a purely descriptive level, there is no attempt to prove whether these changes are causally linked to the well known (but here not described) degenerative changes typical for aging nerves. Consequently, the data presented do not allow to discriminate whether the molecular changes are cause or consequence of degenerative processes within the aging nerve. For instance, it is conceivable that degenerative features could hypothetically be initiated by senescent neurons and axons which could – like a Wallerian-like process - lead to the here observed changes, thus being rather the consequence than the primary cause for degeneration. While it is, nevertheless, possible that nerve degeneration may be forced by the molecular changes described here, there is – as said – no attempt to decipher such a secondary amplification mechanism, for instance by targeting the identified molecules and/or the respective cell types or the proposed cell-cell interactions. This is a serious drawback of the study that substantially limits its contribution to understand peripheral nerve degeneration at advanced ages.

We totally agree with the reviewer that this is an extremely interesting point. In our previous work, we detected reduced nerve fiber and Schwann cell density (Sassu et al, 2024) in the sciatic and cardiac nerves 20-24 months old mice. In the new version of the manuscript, we repeated those experiments, including also sciatic nerves from 15-16 months old mice (Fig S1). We found that Schwann cell relative numbers are already

reduced in the sciatic nerve from 15-16 mo old mice. As for TH+ nerve fibers, we found that there might be a reduction already in the 15-16 mo group ($p=0.1017$) statistical significance. This could point to Schwann cells being affected by aging first, but a more detailed project will be necessary to confirm this and establish the mechanisms behind.

B) Major criticism on experimental performance and interpretation:

a) *the presentation of the snRNA-seq data does not follow the state-of-the-art frame: additional genes for the different cellular populations are missing, as well as heat map analysis, differential gene expression analysis, ViolinPlots and FeaturePlots. Based on these omissions, the study is far away from being a “sciatic nerve aging atlas” as the authors claim (see for comparison, Gerber et al., 2021; Yim et al., 2022).*

In the re-submitted version, we have now included a heatmap with top enriched genes for each of the 19 first level clusters (Fig. 1 and Fig. S1). In addition, we have expanded the marker genes for the populations, and included a bar-plot with the relative presence of each cell type.

We have also setup a data portal to explore the snRNA-seq data that will be open to the public upon manuscript acceptance:

[FIGURE REDACTED]

b) *established macrophage markers like Csf1r and Adgre1 are not represented in the ViolinPlots in Fig. 1. CD45 is not a specific marker for macrophages, but a global immunemarker.*

We appreciate the reviewer suggestion and have included the recommended genes. Also, we are aware that CD45 is a marker of all the bone-marrow derived cells, we apologize for not having included further macrophage markers in the original violin plot with the different populations. We hope the new figures are clearer.

c) *LYVE1 is also associated with epi-/perineurial macrophages and not confined to perivascular ones (see Ydens et al., 2020). **AND** e) The M1/M2 nomenclature is old-fashioned and does not represent the recent state of the art. It was previously used at “pre-single cell/nucleus-analysis” times as an approach to try to classify “good” and “bad” macrophages but is outdated at times when more sophisticated methods (like snRNA-seq) are available*

We appreciate the reference as it has made us improve our annotation. We have found indeed that the cluster previously labeled as perivascular macrophages

corresponds, in agreement with the citation, with epicardial macrophages. In addition, we have relabeled the macrophages based on the work by Ydens et al, but also based on their functions, moving away from the M1/M2 nomenclature. We think that this highly improves the quality of our analysis and we are very grateful to the reviewer for the citation.

- d) *the GLUT1+ immunoreactive, more linear profiles in Fig. 2E (middle; “16 m.o.”) do not look like macrophages at all. Rather, they look like blood vessels being in line with the established finding that GLUT1 is a marker for endothelial cells (see also Gerber et al., 2021). In the left and right panel of Fig. 2E, the positive structures look like slim perineurial cells, without any hint for phagocytosis, while phagocytic cells are usually more rounded and not arranged as linear structures. Thus, designating GLUT1 as phagocytic marker does not at all fit to the immunocytochemical staining, corroborating the overall questionable validation approaches.*

We agree with the reviewer that GLUT1 is not a specific marker for phagocytic macrophages. In fact, in the peripheral nerves is also considered a marker of perineurial fibroblasts (which correspond to the lineal structures observed in single positive cells). On the other hand, we combined the bone marrow-derived cell marker CD45 with GLUT1 to separate other GLUT1-positive cells from the macrophages. In addition, we have performed a new experiment combining CD68 as a macrophage marker and fluoromyelin to explore myelin phagocytosis at different age-points. The new results support the finding, with increased numbers of macrophages phagocytosing myelin in the 15-16 months old group. In addition, we provide with relative numbers of CD68+ macrophages, which are in line with those observed for CD45+, We hope this explanation satisfies the reviewer questions about our approach.

- f) *Cspg4+ fibroblast have been found in datasets from other laboratories. However, they are described already in 60d-old mice.*

That is correct. Cspg4 is normally expressed in endoneurial fibroblasts and we observed so in our datasheet. In old mice we observed significantly more deposits of Cspg4+ due to the activation of the chondrocyte-like state in peri- and epi-neurial fibroblasts.

- g) *there is a plethora of literature that describes the pathological and pathophysiological features of aging nerves: **Ceballos et al., 1999**; **Cowen et al., 2005**; **Leblhuber et al., 2011**; **Anish et al., 2015**; **Ward et al., 2015, 2016**; **Canta et al., 2016**; **Moldovan et al., 2016**. Unfortunately, none of the papers indicated is cited, although it would help the reader to understand the aim of the study and would represent a comprehensive understanding of the topic by the authors.*

We appreciate the bibliographical advice. We have now incorporated the fitting citations in the manuscript.

- h) *According to the authors view, FGF2 expression appears to be a pivotal player regarding the development of chondrocyte-like cells, indicative of aging fibroblasts. It is, therefore, essential to verify the expression by qPCR and/or Western blot analysis in a time-dependent manner.*

We have now added the adipocyte FGF2 and FGF1, as well as chondrocyte-like fibroblasts FGFR1 and 2 combined expression counts per animal from snRNA-seq in a new panel (Fig. S4C, D, E) to extend our previous analysis and fluorescence analysis.

- i) *A range of 20–30 months is quite broad. Could the authors clarify the specific ages and the rationale of selection of the ages investigated in the study?*

The rationale behind this analysis was to use mice as old as possible in the final time point. Mice are considered old after 20 months and therefore we established that as the minimum age to be included in that group. On the other hand, we let the mice in this group live up to 30 months old as long as they had quality of life. If suffering was observed, the animals were sacrificed, and their organs harvested. The ages of the animals in the oldest group were:

30 months old	3
28 months old	1
25 months old	2
22 months old	1
20 months old	1

C) Minor, but significant problems:

- a) *When addressing nerve adipocytes, the study by Sundaram et al. (2023) should be considered. It is very well known that adipocyte-like cells are at the edge of the nerve(peri-/epineurium), while it appears that the authors are surprised by that, as they claim: “Remarkably, these adipocytes where (should read: were) detected mainly in the epi- and perineurium, making local adipocyte-hPnFbs interaction feasible (Fig 4L). Importantly, fibroblasts can also be – as opposed to adipocytes – endoneurial.*

This is a fascinating work and we really appreciate the citation recommendation. The text has been modified and this work has been added to the references list:

“During aging, there is also an accumulation of adipocytes and lipids in different organs¹³, and cross-talk between adipocytes and peripheral glial cells have been previously reported³⁵”

Also, while not surprised, we just wanted to remark that adipocytes locate around epi/perinerium. In any case, we have removed the word “remarkably” from the text.

- b) *Introduction: what are “circulating phagocytic macrophages”?*

We apologize for the confusing nomenclature. We meant circulating (Ccr2+/Cd74+/Hsp5+)-derived phagocytic macrophages, in contrast to resident (Cx3cr1+) macrophages.

- c) *Fig. 6: the schematic drawing of the nerve does not fit with the nerve architecture in mice. Rather, the subdivision into individual fascicles is typical for human nerves; maybe a scheme of a human nerve may have been erroneously taken as a base. Furthermore, in the schematic summary it looks like as FGF2 is derived from senescent fibroblasts.*

The schematic drawing has now been changed into a mouse-representative sciatic nerve.

- d) *Fig. 1a: the nerve isolation scheme does not represent the position of the sciatic nerve.*

In the schematic it is shown from the point of view the access used to harvest the sciatic nerve, in supine position. We have modified the diagram to make it clearer.

- e) *How do the authors conclude that Tgfb1 (Transforming growth factor beta receptor 1) is a specific marker for macrophage activation in the PNS? Is there specific literature available? Moreover, in Table 1, the authors erroneously name Tgf... as "tissue growth factor...".*

We apologize for the mistake, now it reads "Transforming growth factor". Also, as mentioned before, we have re-evaluated the macrophage clusters and renamed them based on more recent findings. The new name for the TGFBR1+ cluster is "chronic inflammation macrophage" in agreement with the work by Zhigang Lei et al. 2024. *JCI insight*.

Response to reviewer #2

We appreciate the reviewers help on the revision process.

Response to reviewer #3

Major points:

1. *The manuscript currently reports an interesting finding but remains fully descriptive although many straight-forward follow-up experiments could be envisioned. What exact lyis the functional relevance of the proposed chondrocyte-like transcriptional state induced by FGF2? Does this directly or indirectly the function of the peripheral nerves? For example, does this impair nerve conduction or propagate axonal loss or myelin loss in ageing?*

We agree with all the points that reviewer makes here. This is an initial work that will be followed up by more studies trying to clarify points like the ultimate role of the chondrocyte-like state activation of neural fibroblasts, and the increased CSPG4 deposition.

Would FGF2 overexpression or addition induce similar changes and then functional impairments without ageing. Or would such chondrocyte-like cellular states alter the stiffness / flexibility of the nerves? Or would FGF2 activation induce endoneurial or perineurial fibrosis or swelling or cellular proliferation?

In our *in vitro* model, senescence was required for FGF2 to induce the chondrocyte-like state, so we think that aging is required for the chondrocyte-like state activation via FGF2. In addition, we did not observe changes in proliferation in healthy or senescent hPnFbs upon FGF2 treatment (Fig S4 and Fig 5).

As it stands, Fig 3H shows relative proportions but now change in cell numbers or nerve area etc.

As the reviewer correctly comments, Fig 3H shows the relative numbers of perineurial or epineurial fibroblasts that express SOX9, but in Fig 3F a significant increase in the number of SOX9+ cells were detected.

- 2. And if the answer to any of the prior questions is yes, would deficiency of FGF2 or of its down-stream signaling pathways in nerve fibroblasts then abrogate age-induced pathology? An FGFR-conditional mutant is available*

As mentioned, we plan to follow up this work with *in vivo* mechanistical studies to better understand aging of peripheral nerve cells and how to intervene it to slow down deterioration. We appreciate a lot the information about the FGFR conditional mouse, it will be very handy in our future projects.

- 3. An alternative or additional way of increasing the relevance of the finding could be to aim for human confirmation. Observing similar alterations in peripheral nerves (e.g. suralnerve biopsies) of human individuals with polyneuropathies vs. 'non-diseased' ageing vs. young individuals would support the relevance of the findings and improve credibility.*

We fully agree with this, and we have performed new experiments with the support of Prof. Dr. Marco Prinz and Dr. Lauritz Miarka. In the new experiments, we compare FGF2 and SOX9 expression in peripheral nerves from younger and older donors, finding that in all the cases, the samples from older donors have more SOX9+ and SOX9+/FGF2+ cells, and higher FGF2 levels. We appreciated the input, as our new findings increase the translational relevance of our findings.

- 4. With their simplified M1 vs M2 macrophage phenotyping, the authors omit important parts of the literature demonstrating the existence of endo- vs. epineurial macrophages with distinct transcriptional phenotype and distinct ontogenetic origin (DOIs:10.1038/s41467-020-16355-w, 10.1038/s41593-020-0618-6,10.1073/pnas.1912139117). The authors should also cite other papers than Yim et al 2022 Nat Neurosci reporting scRNA-seq characterization of peripheral nerves (e.g. DOI10.7554/eLife.58591)*

We agree with all the reviewers that our work on macrophage characterization was not ideal. For this reason, we went back to our data and based on the bibliography suggested we re-analyzed our macrophage cluster. We have annotated epi and endoneurial resident macrophages and annotated the rest based on their possible origin (circulating monocytes vs resident macrophages) and their function.

We have also commented on and incorporated the citation for the work by Gerber et al (eLife 2021):

“While in the last 10 years this tool has helped to better understand the heterocellular composition of the peripheral nerves^{24,25}, and their cellular changes after nerve injury²⁰, we still lack a cell atlas of peripheral nerve aging that helps us understand age-related changes of expression in the PNS.”

Minor points:

1. *Labels and font sizes in Fig 2A-D should be bigger*

We have increased the font size in the new version of the figure.

2. *Fig 2F is a sub-optimal way of depicting this. Consider depicting proportions of total cells (e.g. in stacked barplots) rather than individual cells.*

Fig 2F represents proportions of each subtype of macrophage in relation to the total number of macrophages. Now the percentage of each subpopulation has been included in the stacked barplots.

Response to reviewer #4

1. *Introduction: The description of SOX9 as a "marker" for endoneurial fibroblasts and satellite cells during repair could be clarified. The term "marker" implies specificity, which may not be accurate. A more precise phrase, such as such as “SOX9 is expressed in these cell types under specific conditions,” would be preferable.*

We agree with the reviewer and now the text has been modified accordingly:

“In peripheral nerves from young adults, SRY-box transcription factor 9 (SOX9) has been found to be expressed in endoneurial fibroblasts¹⁷ “

2. *Figure 1D: The data suggest a reduced presence of B cells in aged nerves, but additional evidence is needed. An extra panel in Fig. S1 showing the nuclei density of the B-cell population could clarify this. Additionally, a shift in the EC1 population at 16 months is noted but not discussed. It would be useful to include a brief comment on this observation, as the shift in macrophage populations is already addressed.*

This is an interesting point and now we have included analyses for ECs and B-cells in the Fig S1. While uniquely based on the UMAP one might see some differences, we did not find statistically significant differences in relative numbers of any of the mentioned clusters.

3. *Inflammaging: While inflammaging and macrophage activation are discussed, the addition of a reference such as Büttner et al. (2018), which describes an increase in macrophages in aging nerves, would strengthen the argument.*

The citation and concept have been incorporated in the discussion:

“In addition, previous studies have found increased macrophage numbers in the sciatic nerve of old mice²⁸, and that a reduction in macrophage numbers also reduces nerve structural and functional impairment associated with aging²⁹.”

4. *Macrophage Phagocytosis: GLUT1 is used as a marker for phagocytic activity in macrophages, with increased phagocytosis noted during aging. It would be valuable to test whether macrophages are phagocytosing myelin, perhaps using F4/80 and Fluoromyelinco-staining in sciatic nerve sections. Since abnormal myelin is mentioned in the discussion, additional experimental evidence here would be beneficial.*

We appreciate the input and we have run the suggested experiment. We used CD68 as a macrophage marker combined with fluoromyelin, finding increased phagocytosis starting at 15-16 months old, which builds on our previous data. Thank you for the recommendation, as this experiment strengthen our results.

5. *5. Figure 2E: Phagocytosis is increased in middle-aged animals but not in old animals. This discrepancy should be discussed. One possible explanation is that macrophage dysfunction may occur with aging, preventing effective phagocytosis. This hypothesis should be explored further.*

We appreciate the suggestion; this idea is now included in the discussion:

Finally, it is important to remark that we observed a reduced number of phagocytic macrophages in the nerves from the oldest mice. This could be originated by an age-associated increase of macrophage malfunction, as it has been previously reported⁴⁰.

6. *Fibrosis and Calcification: The development of fibrosis and calcification in organs such as the heart and lung is well-documented during aging. A reference supporting this in the context of peripheral nerve aging would be useful.*

Fibrosis and calcification in the cardiovascular system (and others) are indeed well documented and we have included a citation in this regard (#12). In regard to collagen and proteoglycan deposition in aged nerves we included three citations (#15, #36, #37).

7. *Collagen Expression: The increased expression of Col1a1 and Col2a1 in the aging nerve could be validated through immunofluorescence staining to further support the idea of ECM remodeling. This would provide additional insight into the spatial distribution of these collagens during aging.*

We totally agree with the point raised by the reviewer. We have recently used a fluorescently labelled collagen hybridizing peptide in cardiac and sciatic nerves from old animals where we performed lineage tracing of SC using Sox10CreERT2. We detected much more collagen remodeling in the sciatic nerves from old mice, supporting our snRNAseq findings. Unfortunately, this data is under revision in a different manuscript. In any case, we have included the panel in review for the reviewer.

[FIGURE REDACTED]

8. *Figure 2: The concentration-dependent increase in the chondrocyte-like fibroblast state is an important finding and should be explicitly mentioned in the text.*

We totally agree that this is a very important point, and we think that only due to increased expression of FGF2 in adipocytes combined with increased number of adipocytes could provide with enough signal for this fibroblast activation to happen. In the text it reads:

“Together this data indicates that high concentrations of FGF2 can activate the co-expression of FOXC2 and SOX9, but other signals might be behind CSPG4 increased expression in senescent hPnFbs.”

“Therefore, to better mimic our findings *in vivo*, we decided to compare the effect of low concentrations of FGF1 in combination with high concentrations of FGF2 on hPnFbs (Fig. S8A). Low concentrations of FGF1 were insufficient to block the FGF2 effects, with significantly more SOX9+ (Fig. S8A') and SOX9+/FOXC2+ (Fig. S8A'') hPnFbs after treatment. On the other hand, no change in CSPG4 expression was detected (Fig. S8B, B'). Together this shows that in a more similar environment to what is observed during aging *in vivo* FGF2 can activate co-expression of SOX9 and FOXC2, but higher FGF1 concentrations can block activation.”

9. *FGF2 Expression: The expression of Fgf2 could be further contextualized by adding fold-change or ratio data for the three timepoints, which would strengthen the physiological relevance of the findings and justify the use of high concentrations of Fgf2 in the cell culture model.*

We appreciate this great idea. Now the raw counts for FGF2 and FGF1 can be found in Fig S4C and D, respectively.

10. *Graphical Abstract: The reversibility of the chondrocyte-like state, demonstrated by the ability of Fgf1 (at sufficient concentrations) to revert the transformation, should be indicated in the graphical abstract. Adding a dashed line or other visual cue in the bottom panel could highlight this potential therapeutic avenue.*

We appreciate the comment, we have now added the requested change to Fig6.

11. *Methods Section: The number of animals used to isolate enough nuclei for sequencing is not specified. This should be included for transparency.*

Both sciatic nerves from a single mouse were used. This can be found in the “Nuclei isolation and single nucleus RNA-seq (snRNA-seq) section of methods methods:

“From each mouse, both snap frozen femoral sciatic nerves were thawed in nuclei isolation media (NIM, 250 mM Sucrose, 25 mM KCl, 5 mM MgCl₂, 10 mM Tris-HCl) with 25 mM DTT, 1X protease inhibitor (05056489001, Roche), 0.1% Triton X-100 and RNase inhibitor (N2515, Promega). Each pair of nerves was homogenized using a Kimble dounce tissue grinder (D8938, Merck) and the resulting suspension was filtered through a 30 µm CellTRics (Sysmex, MSA150914) into a LoBind Eppendorf tube (Eppendorf). Nuclei were pelleted (1000 x g, 10 min, 4°C) and washed with NIM + DTT + Protease inhibitor (05056489001, Roche) and RNase inhibitor (N2515, Promega). After the wash, nuclei were resuspended in sorting buffer (1 mM EDTA, 0.2 U/µL RNase inhibitor (N2515, Promega), 2% fatty acid-free BSA (A7030-100g, Sigma) in PBS). Finally, the nuclei were filtered again through a 30 µm CellTRics (Sysmex, MSA150914), and DRAQ7 (#7406, Cell Signalling) was added to the nuclei suspension for nuclei detection by flow cytometry. 10,000-20,000 nuclei were sorted using a S3 Fluorescence-activated cell sorter (Bio-Rad).”

12. *Aging and Early Changes: The data suggest that age-related changes in the PNS begins early as 15-16 months. The discussion could benefit from further exploration of what drives these early changes and their implications for later-stage aging.*

We think this is a very interesting point, and we have included a paragraph discussing early vs late aging changes in peripheral nerves at the end.

“Our data also indicates that aging changes in the peripheral nerve start as early as at 15-16 months of age. Nerve fiber reduction and possibly SC numbers are already down, while increased FGF2+ adipocytes and increased gene expression of chondral markers in neural fibroblasts are already detected in the SN of 15-16 months old mice. On the other hand, increased deposition of CSPG4 was only detected in the 20-30 months old age group. This points to CSPG4 deposition being a consequence of other initial aging alterations, and makes futures studies of the effect of FGF2 on SC and nerve fibers, relevant. “

Figure Labeling and Presentation:

We have fixed the labeling and presentation as requested.

Sincerely,

Luis Hortells
Ass. Professor

Sebastian Preissl
Professor

16.07.2025

MS# NCOMMS-24-68816

Dear colleagues,

We would like to thank you for the constructive comments and consideration of our manuscript "FGF signaling induces a chondrocyte-like state of peripheral nerve fibroblast during aging". To answer the reviewers comments and suggestions, we have performed additional experiments to further corroborate our findings, further analysed our snRNAseq data, and added important points to the discussion.

Specific comments from each reviewer have been addressed and described below. Specific changes/additions to the manuscript have been highlighted (in yellow) in the markup version of the text.

We believe that the revised manuscript addresses the reviewers' comments, which have helped to improve quality and relevance of the paper.

Thank you.

Reviewer #2

"a) Despite some attempts in the revised manuscript, the data still does not allow for a clear distinction between whether the molecular changes are a cause or a consequence of degenerative processes within the aging nerve. Further clarification or additional evidence would be necessary to address this issue. For example, if histopathological changes in the peripheral nerve occur earlier (see also comment b), but CSPG4 deposition and FGF expression are observed at later stages, could these be rather considered as a consequence of degeneration? Additionally, the statement that "both SOX9+ cell density and CSPG4 area were significantly increased starting from the 15-month time point (Fig. 3F and F)" is confusing, as the quantification does not support this. The significance was only observed at the 20-24 month time point, and this discrepancy should be addressed."

We agree that further research on peripheral nerve aging is required to clarify exactly what changes are triggers and what consequences. With the results obtained in our work, we have further discussed the order of events that we have identified. While the earlier detection of the triad of glial cell loss, nerve fiber loss, and increased macrophage phagocytic activity seems to point to degeneration of the nerve unit as the initial issue, increased expression of FGF2 is already observed in the nerves. In our opinion, degeneration, initiated by Schwann cell malfunction is the "origin of all problems", but further experiments will be required to answer this question and explore the molecular mechanisms behind this. Finally, we apologize for the confusion, we have substituted 15-month for 20-month time point.

“b) The new S100B staining presented in the manuscript (Figure S1A) does not appear convincing compared to existing literature (e.g., DOI: 10.1371/journal.pone.0123278, DOI: 10.3390/ijms18020263). It may be beneficial to include additional markers, such as Sox10, as previously performed by the authors, to better quantify Schwann cell numbers. Furthermore, quantifying the relative numbers of S100B+ cells per total DAPI+ nuclei does not fully capture a loss of Schwann cells, as it may be influenced by an increase in the number of other endoneurial cells (such as macrophages and fibroblasts) in aging nerves. The authors also state that two defined hallmarks of peripheral nerve aging are the loss of nerve fibers and the loss of Schwann cells. However, other studies have demonstrated additional pathological changes associated with aging, including demyelination, Schwann cell proliferation and onion bulb formation. These features are also important hallmarks of age-related peripheral nerve pathology and should be acknowledged to provide a more comprehensive view. Therefore, the conclusion that “aging changes in the peripheral nerve start as early as at 15–16 months of age” requires further validation. Supporting evidence such as neurofilament (NF) and P0/MBP staining on cross-sections (see also DOI: 10.12659/MSM.918277) or ultrastructural analyses using high-resolution electron microscopy (EM) would strengthen the hypothesis.”

Based on the suggestion by the reviewer we have performed SOX10 IF (Fig S.1A, A”). Interestingly, SOX10+ Schwann cell presence seem to be significantly down only in the oldest mice (20-30 m.o.). To interrogate the differences between S100B+ and SOX10+ cell numbers, we explored whether both markers are expressed equally in all the subpopulations of Schwann cells, and found that S100B is mainly expressed in the myelinating Schwann cell populations (Fig. S1B, C, D). Therefore, we think that it is possible that the reduction in Schwann cell numbers affects first to the myelinating Schwann cells and later in time to all Schwann cells.

We agree with the reviewer that abnormal myelin is an important hallmark of aging. Thus, we have performed additional analysis of fluoromyelin in the nerves (Fig S1E, E”). Using fluoromyelin we have not found significant differences in myelin area in the nerves, but we are aware this analysis might not be able to detect fine age-associated myelin abnormalities as good as electron microscopy, as other groups have reported.

“c) The authors present new data on FGF expression counts per animal based on the snRNA-seq approach. However, as FGF2 expression appears to be the pivotal player in the development of chondrocyte-like cells, it is essential to verify the expression by another approach (qPCR and/or Western blot analysis) in a time-dependent manner.”

We agree that results from snRNAseq need confirmation. Validation of scRNA-seq data using qPCR or Western Blot in chondrocyte-like cells and adipocytes is a great suggestion. However, chondrocyte-like cells and adipocytes are not the main cell types in the nerves and their fractions are dynamic during aging. Thus, this analysis would require nerves from additional mice from all time points (3-24 months old) and establishing new protocols for purification of these cell populations. On the tissue level, changes in gene expression could be confounded by the changes in cellular composition. Unfortunately, we do not have biosamples to be able to perform these experiments in a reasonable time frame for this revision.

However, our current data provide additional evidence for the role of FGF2 signaling predicted from snRNA-seq. First, we have validated FGF2 protein presence in adipocytes using immunofluorescence and could show higher fraction of FGF2 positive adipocytes (Fig 4L). In addition, our in vitro results show that stimulation of fibroblasts induces a chondrocyte like cell state with increased protein levels of SOX9 and FOXC2.

“d) The immunohistochemical expression of SOX9 and CSPG4 should be validated with a pan-fibroblast marker to confirm the fibroblast identity of the proposed chondrocyte-like cells (Figure 3). Furthermore, if the authors suggest that the most probable origin of these chondrocyte-like fibroblasts is the perineurial fibroblast (PnFb) population, and that endoneurial fibroblasts are the least likely source, this raises an important question: how is the age-dependent signaling mediated within the endoneurium - the primary site of nerve damage? Do the authors propose that fibroblasts migrate from the perineurium to the endoneurium? At present, the manuscript

does not really provide direct evidence supporting that the chondrocyte-like fibroblast state most likely originates from PnFbs in the sciatic nerve of aged mice. Clarification and additional validation would be important to support this conclusion.”

Initially, we explored the most likely cell of origin for the chondrocyte-like state fibroblasts via pseudotime analysis (Fig 3G). This analysis predicted perineurial cells as the most likely cell type that could generate the chondrocyte-like cells, with epineurial being second. Endoneurial were the least likely origin. For this reason, we used well established markers for epi- and peri-neurial fibroblasts (Yim, A. K. et al. Nat Neurosci, 2022). The reviewer suggests the use of a “panfibroblast marker” but to the best of our knowledge, there is not a marker that is exclusively expressed by fibroblast as vimentin is expressed by other interstitial cells like endothelial cells and collagens are expressed by different cell types as well, like nmSC. We agree that further study endoneurial fibroblast aging would be an interesting topic for a follow up work. Regarding migration, while proving cell migration in vivo is extremely challenging, we think that this is a real possibility, as we find perineurial and epineurial cells expressing SOX9 in endoneurial areas.

“e) In Figure 3H, the SOX9/KCN2 immunoreactive profiles (fibroblasts) are indicated by red arrows. However, these profiles are detectable in the endoneurium, rather than the epineurial layer as indicated in the quantification. This discrepancy should be clarified to ensure accurate interpretation of the data.”

As mentioned before, migration is indeed, and interesting possibility and we have included it now in the discussion.

“f) Regarding the human data presented in Figure 5I: could the authors clarify whether the observations are based on comparable peripheral nerves, and whether differences in nerve type may influence their findings? Why are less nuclei present in the nerve biopsy (Human -Old)? To more accurately confirm and validate the specific cell populations that express FGF2 and SOX9, co-staining with established fibroblast and adipocyte markers would strengthen the conclusions. The presence of FGF2-expressing adipocytes within the nerve endoneurium is unusual (Figure 5I). Could the authors provide an explanation or relevant literature to support this observation? Furthermore, in the mouse dataset, SOX9+ fibroblasts and FGF2+ adipocytes are shown as two independent populations. Why are these markers quantified together in the human samples (SOX9+ / FGF2+ cells; Figure 5I”)?”

We agree with the reviewer. We have now included new text in the discussion acknowledging the variable origin of human nerves as a weakness of our work. Regarding the nuclei density, we have now measured it and a reduction in nuclei density is possible, supporting our previous findings in sciatic nerves from mice (Sasu, E et al, 2024).

We also included a new comment on the very interesting FGF2-SOX9 co-expressing cells. We think that in humans, for unknown reasons, the possible chondrocyte-like cells might increase FGF2 expression as a positive-feed loop that maintains the cells and their neighbors in this state. This might be due to the longer life expectancy of humans but also due to pure dimensions of the nerves and the lack of input from adipocytes for longer time. Also, fatty infiltration of peripheral nerves is described in some diseases like lipomatosis of nerve, so the possibility of adipocyte presence in the endoneurium would not be unprecedented. In the case of aging might not be severe enough for the pathologist to detect it. Also, given the fatty nature of myelin, traditional staining techniques like oil red would not differentiate the presence of a small number of adipocytes infiltrating the nerve.

“g) Regarding the validation of phagocytosing macrophages using GLUT1 immunohistochemistry: as mentioned in the previous review, the observed profiles resemble blood vessels more than phagocytosing macrophages. GLUT1 is known to be expressed in endothelial cells (see Gerber et al., 2021), and there does not appear to be a complete overlap between CD45 and GLUT1 staining. Additionally, similar GLUT1-immunoreactive profiles in Figure 3H are interpreted as perineurial fibroblasts. Could the authors clarify the rationale for using GLUT1, especially given that this marker is neither present nor indicated in the phagocytic

macrophage cluster (defined by Cd44, C1qa and Grn)? Based on this, GLUT1 does not appear to be a reliable marker for identifying phagocytosing macrophages and its use in this context should be reconsidered.”

We are aware that GLUT1 is not exclusively expressed in one cell type, but in the case of macrophages, we argue that increased numbers of macrophages expressing GLUT1 is compatible with increased numbers of phagocytic macrophages. To differentiate among cell types we used co-stains, CD45 for macrophages (bone marrow lineage) and SOX9 for chondrocyte-like cells. In the previous version of the manuscript, we further confirmed the presence of phagocytic macrophages phagocytosing myelin using a CD68/fluoromyelin co-stain. To confirm the usefulness of GLUT1, we have incorporated in the new version a CD68/fluoromyelin/GLUT1 co-stain. The results show co-expression of the macrophage marker CD68, together with GLUT1, in cells with phagocytosed myelin (Fig. S3H).

“h) Several of the nerve sections lack optimal quality, as samples are not perfectly longitudinally oriented (e.g., Figure 2F), and some display structural disruption (e.g., Figure S3H). This is not state-of-the art and complicates interpretation. Improving section quality would enhance the reliability of the histological analysis.”

We have modified the suggested panels.

“Minor:

a) Wording: “We detected higher numbers of macrophages with intracellular myelin was found in sciatic nerves from...”. (was found should be omitted)”

b) While the authors mention moving away from the M1/M2 nomenclature, several parts of the manuscript (e.g., Figure S3, Fig 6) still contain the M1/M2 terminology. For consistency, revising the manuscript accordingly is recommended.”

We apologize for both errors, they are fixed now.

c) It would be helpful to include the relative percentages of fibroblast subtypes across ages (similar to how macrophage subclusters are presented in Figure 2).

We agree with this, and this information is now included in the extended information of the manuscript.

“d) Could the authors clarify why tyrosine hydroxylase (TH), which is particularly expressed in peripheral sympathetic neurons, was used as a marker for PNS axons in Figure S1? It would be helpful to understand the rationale for choosing TH over more commonly used axonal markers in the sciatic nerve, such as neurofilament.”

General loss of nerve fibers associated with aging has been previously reported by other groups. Our group has been working with the autonomic nervous system, and we are particularly interested in sympathetic and parasympathetic nerve fibers. We picked TH+ fibers to build on previous general nerve fiber loss reports.

“e) For clarity and transparency, the age range (specific ages and rationale) of the animals used in this study (20-30 m.o.) should be explicitly stated in the Materials and Methods section.”

We agree with the reviewer. This information has been now included in the M&M section of the manuscript.

“f) The authors state that “...we detected a previously undescribed population of fibroblasts that expressed a combination of chondrocyte-lineage markers (Sox9, Foxc2, and Cspg4)...”. However, as noted in the previous review, the expression of these genes is detectable in 60-day-

old mice (see <https://snat.ethz.ch/index.html>). It would be helpful to clarify how this population is considered novel in light of existing data.”

In our dataset due to the higher number of Sox9/Foxc2/Cspg4 positive cells in the nerves from old mice, they formed their own independent cluster. This might have been a reason why this population was not specifically mentioned before.

We have changed the wording to reflect this better:

“In addition, we found an age-related expansion of a fibroblast subpopulation that expressed a combination of chondrocyte-lineage markers (Sox9, Foxc2, and Cspg4). Re-analysis of a published dataset showed presence of this subpopulation in nerves from P60 mice as sub-part of the perineurial fibroblast cluster”

“g) The authors mention that the “chronic inflammation macrophage” cluster (Tgfbr1+) aligns with the findings of Zhigang Lei et al., 2024 (JCI Insight). However, this referenced paper primarily describes macrophages in the context of cancer or chronic infection, not in PNS tissue. Is there other literature available demonstrating Tgfbr1 expression as a marker for macrophage activation in the PNS? Additionally, this referenced paper focuses on PD-1 expression on macrophages in “chronic inflammatory tissues”, rather than specifically addressing “chronic inflammatory macrophages”. Lastly, are similar markers expressed in the Tgfbr1+ macrophage cluster in the PNS compared to the cancer macrophage subtype described in the referenced paper? For instance, does this cluster show high expression of Pcdcd1 or other related markers?”

Ageing is associated with inflammaging, a chronic inflammatory state, and therefore, it is not completely surprising that there are macrophages with features of other chronic inflammatory processes. We have checked the presence of other cancer-associated markers, and we have not found similar markers in the genes that we explored. This possibly indicates that while chronic, not all the inflammatory processes and macrophages are equal.

“h) Figure S2H (middle panel): does the strongly reduced Fluoromyelin intensity indicate overall myelin loss/demyelination in 16 m.o. sciatic nerves or is this signal loss due to technical reasons?”

We have now measured the fluoromyelin area and there is not a significant difference, so we assumed that this was technical artifact. We have changed to a more representative image.

Reviewer #3 (Remarks to the Author):

“My previous concern with the manuscript was the descriptiveness of the data since no functional relevance of the FGF2 signalling pathway was provided. The authors do to not provide any additional data in this regard.”

Unfortunately, genetically modified mouse work would require several years of work, but we performed the in vitro studies with human cells in order to provide some molecular mechanisms, testing at the same time possible relevance in humans.

“Instead, the authors now provide FGF2 / SOX9 staining of human peripheral nerve specimen from 2 young and 3 aged individuals and depict representative stainings confirming their key observation in humans in vivo. This does improve the credibility of the finding. However, the human data remain limited in several aspects:

a) In the online methods text, I do not find information on how the proportion of FGF2/SOX9 stained cells was quantified. Is this stained cells per all DAPI+ cells?”

We are sorry if this was not clear, as figure 5 I', I'', and I''' indicate, the normalization was done, indeed, compared to number of cells (DAPI+).

“b) If this (apparently) is cell density, how would absolute number look?”

The raw numbers of FGF2 and FGF2/SOX9 cells follow the same pattern as when normalized by nuclei. On the other hand, density of SOX9+ cells is increased, while raw numbers are more stable.

This has led us to quantify the density of cells in the human nerves, and we observed that indeed nerves from old humans seem to replicate a reduction in cell density as it was reported in mice (Sassu et al 2024). We appreciate the reviewers comment as this is important data, that has been now included in the manuscript and the discussion.

“c) I consider a 2 vs 3 comparison without testing for significance inadequate.“
We completely understand the concerns of the reviewer. We have avoided to make any full claim and just point to the directionality of the data to prevent misinterpretations but unfortunately, we have not been able to locate more healthy samples of motoric and mixed nerves.

“d) It remains debatable how well mixed nerves (e.g. ulnar nerve) can be compared with purely motor nerves (e.g. hypoglossus, accessory nerve) as outlined in the methods; especially given that the Schwann cell phenotype (and myelin protein composition) differs between motor and sensory nerves / fibres (e.g. Yim 2022 Fig 5) „

We agree with the reviewer that this needs to be commented and discussed. We have done so in the new version, acknowledging variability in the samples as a potential weakness of the study.

Sincerely,

Luis Hortells

Sebastian Preissl

18.08.2025

MS# NCOMMS-24-68816

Dear colleagues,

We would like to thank you for the comments and suggestions of our manuscript “FGF signaling induces a chondrocyte-like state of peripheral nerve fibroblast during aging”. To answer the reviewers’ comments, we have removed confusing experiments, performed additional experiments to strengthen our findings, reanalyzed data as requested and included human sensory nerve samples to be able to perform statistical analysis.

Specific comments from each reviewer have been addressed and described below. Specific changes/additions to the manuscript have been highlighted (in yellow) in the markup version of the text.

We believe that the revised manuscript addresses the reviewers’ comments, which have helped to strengthen our work.

Thank you.

Reviewer #2

“-the referee is still surprised that the authors use GLUT1 as a marker for phagocytosing macrophages. Slc2a1 (Glut1) is not detectable in the macrophage clusters (see Violin plots in Figure 1e) but is present in endothelial cells (EC) and pericytes. This may also account for the blood vessel-like morphology observed in the immunohistochemical stainings (see previous comments to the authors). It would be helpful to display the separated staining channels in Figure S3H to clarify whether macrophages are simply localized near blood vessels, as frequently observed in high-resolution electron microscopy studies and as suggested by the current staining, or really express Glut1. Co-staining with an additional blood vessel marker would also be advantageous.”

To avoid confusion and improve clarity we have removed the GLUT1 macrophage experiments and performed stainings using GRN, a phagocytic marker which is also highly expressed by this cluster. The results show a progressive increase in phagocytic macrophages that was also observed when counting myelin-containing macrophages containing.

“as previously noted, to confirm and validate the specific cell populations expressing FGF2 and SOX9, co-staining with well-established fibroblast markers would strengthen the conclusions. Pdgfra or Cd34 are recommended markers for fibroblasts.”

In this case we have to re-iterate that to the date, no pan-fibroblast markers are known in nerves or other tissues. We have used markers of peri- and epineurial fibroblast combined with SOX9. The suggested markers (PDGFRa and CD34) are, unfortunately, expressed by many cell types, with low expression on the fibroblasts clusters, with the exception of endoneurial fibroblasts as can be observed in the following UMAPs:

“The previous comment to the authors stated: “In Figure 3H, the SOX9/KCN2 immunoreactive profiles (fibroblasts) are indicated by red arrows. However, these profiles are located in the endoneurium rather than in the epineurial layer, as indicated in the quantification. This discrepancy should be clarified to ensure accurate interpretation of the data.” The authors responded: “As mentioned before, migration is indeed an interesting possibility and we have included it now in the discussion.” Unfortunately, this response does not address the concern regarding the accurate interpretation of the data. To distinguish between epi-, peri-, and endoneurial cells, different arrowhead colors should be used in the representative staining, and the SOX9/KCN2 immunoreactive profiles within the endoneurium should also be quantified and presented.”

To improve clarity, we have removed the arrows pointing to KCNC2+/SOX9+ cells located in the endoneurium (Fig 3H). In addition, we counted the endoneurial KCNC2+/SOX9+ cells and removed them from the total KCNC2+/SOX9+. Consequently, a new graph was included (Fig 3H')

Reviewer #3:

“Unfortunately, however, the limited number of human samples remains unaddressed. My previous wording here was: 'I consider a 2 vs 3 comparison without testing for significance inadequate'

The authors' attempt at discussing this remains disappointing since my concern would have been straightforward to address. Although sural nerve biopsies are performed rarely, every neuropathology department will hold sample banks of human sural nerve biopsy sections across a wide age range.

Increasing the number of stained samples in Fig 5I would thus have been easy and I am still convinced that this would improve the manuscript.”

We agree with the reviewer that being unable to perform statistical analysis was disappointing. Originally, we tried to use only mixed nerves (as for mouse we were using the sciatic nerve), not including pure sensory nerves like the suralis. We now include suralis samples and performed statistical analysis (n=5 in total per age group). FGF2+ cell numbers and FGF2+/SOX9+ cells were significantly increased in older nerves, SOX9+ cells was not statistically significantly, but showed a trend.

Sincerely,

Luis Hortells

Sebastian Preissl